# OUSAC: Optimized gUidance Scheduling with Adaptive Caching for DiT Acceleration

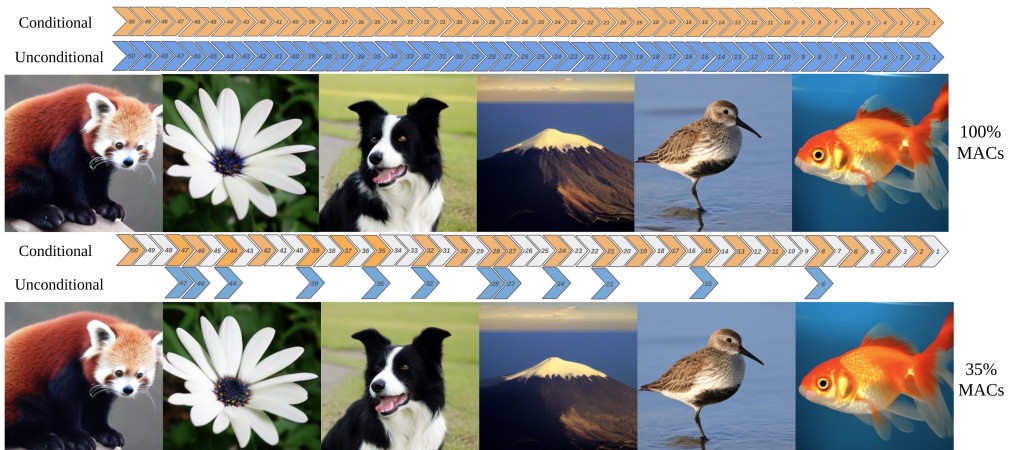

Figure 1: Top: Standard DDIM with 50 steps with CFG uniformly across all timesteps, requiring both conditional (orange) and unconditional (blue) forward pass at each step. Bottom: OUSAC achieves a similar visual quality with only **35%** of the original computational cost via two-stage optimization: optimized sparse guidance scheduling (empty white space) with caching (gray).

## Abstract

Diffusion models have emerged as the dominant paradigm for high-quality image generation, yet their computational expense remains substantial due to iterative denoising. Classifier-Free Guidance (CFG) significantly enhances generation quality and controllability but doubles the computation by requiring both conditional and unconditional forward passes at every timestep. We present OUSAC (**O**ptimized g**U**idance **S**cheduling with **A**daptive **C**aching), a framework that accelerates diffusion transformers (DiT) through systematic optimization. Our key insight is that variable guidance scales enable sparse computation: adjusting scales at certain timesteps can compensate for skipping CFG at others, enabling both fewer total sampling steps and fewer CFG steps while maintaining quality. However, variable guidance patterns introduce denoising deviations that undermine standard caching methods, which assume constant CFG scales across steps. Moreover, different transformer blocks are affected at different levels under dynamic conditions. This paper develops a two-stage approach leveraging these insights. Stage-1 employs evolutionary algorithms to jointly optimize which timesteps to skip and what guidance scale to use, eliminating up to 82% of unconditional passes. Stage-2 introduces adaptive rank allocation that tailors calibration efforts per transformer block, maintaining caching effectiveness under variable guidance. Experiments demonstrate that OUSAC significantly outperforms state-of-the-art acceleration methods, achieving 53% computational savings with 15% quality improvement on DiT-XL/2 (ImageNet 512×512), 60% savings with 16.1% improvement on PixArt-$\alpha$ (MSCOCO), and 5× speedup on FLUX while improving CLIP Score over the 50-step baseline.

# 1 INTRODUCTION

Diffusion models (Ho et al., 2020; Song et al., 2021; Dhariwal & Nichol, 2021; Peebles & Xie, 2023; Chen et al., 2023) have revolutionized generative modeling, achieving unprecedented quality in image synthesis. Yet their widespread adoption remains limited by computational demands: generating a single high-quality image requires trillions of floating-point operations (TeraFLOPs) due to iterative denoising. This cost even doubles when using Classifier-Free Guidance (CFG) (Ho & Salimans, 2022), which improves generation quality by interpolating between conditional and unconditional predictions to strengthen adherence to input conditions. Though CFG is essential for balancing sample diversity and conditional fidelity, it applies a constant guidance scale uniformly across all timesteps – ignoring whether each step equally benefits from guidance.

Existing CFG-related research pursues two distinct objectives. **Efficiency-focused methods** (Castillo et al., 2023; Yuan et al., 2024; Lv et al., 2024) detect when conditional and unconditional outputs are similar to skip redundant computation. However, they keep guidance scales *fixed* and total sampling steps *unchanged*, achieving modest speedups while often degrading quality. **Quality-focused methods** (Gao et al.; Malarz et al., 2025a; Zhu et al., 2025; Sadat et al., 2025) demonstrate that time-varying guidance scales can improve generation quality. Yet they still require full CFG computation at every timestep, providing no computational savings, and their schedules remain "largely heuristic" (Gao et al.). These directions appear incompatible: one reduces computation at the cost of quality, the other improves quality without addressing efficiency.

This raises a fundamental question: *can we achieve both by jointly optimizing when to apply CFG and what guidance scale to use?* A key insight suggests this is possible: *variable guidance enables sparse computation*. Adjusting scales at certain timesteps can compensate for skipping CFG at others. The challenge is non-trivial. The joint search space of discrete skip patterns and continuous scales is intractable for manual design, and naively skipping guidance at arbitrary timesteps leads to severe quality degradation. As illustrated in Figure 2, only through carefully optimized sparse guidance patterns can match the performance of full CFG while eliminating most computational overhead, indicating that many guidance computations in existing approaches are redundant.

We present OUSAC: Optimized gUidance Scheduling with Adaptive Caching for Diffusion Transformer Acceleration, a framework that addresses an unexplored challenge of integrating guidance scheduling with feature caching to accelerate DiT. We focus on transformers due to their growing adoption in state-of-the-art models (Peebles & Xie, 2023; Chen et al., 2023; Labs, 2024) and their uniform block structure that enables systematic optimization. While existing methods achieve efficiency gains through either guidance scheduling (Gao et al.; Wang et al., 2024; Castillo et al., 2023) or feature caching (Ma et al., 2023; 2024; Zou et al., 2025) in isolation, no method has effectively combined the two.

This integration is challenging because variable guidance patterns violate the key assumption of caching methods – feature similarity across timesteps. To tackle this, OUSAC employs a two-stage optimization approach. In *Stage 1*, we discover sparse guidance schedules that identify which denoising steps truly require CFG and determine the optimal guidance scale at each. This poses a complex discrete-continuous optimization problem, as timestep decisions non-linearly interact throughout the denoising trajectory. Gradient-based optimization cannot apply directly to this due to memory constraints and vanishing gradients over $T$ steps. Instead, we use evolutionary strategies to efficiently explore the guidance space, discovering extremely space patterns that preserve quality while skipping guidance at most timesteps. In *Stage 2*, we introduce adaptive rank allocation for incremental calibration under variable guidance. The sparse schedules from *Stage 1* introduce two types of denosing deviations: guidance scale variations between consecutive steps, and branch switching when alternating between CFG and conditional-only passes. These break the feature consistency assumed by standard caching. We address this by assigning different calibration ranks to different transformer blocks, adapting to their sensitivity to guidance changes, an significant departure from the uniform calibration used in prior methods (Chen et al., 2025).

The synergy between optimized scheduling and adaptive caching enables gains beyond what each technique achieves alone. OUSAC reduces computational cost by 53% while improving FID by 15% on DiT-XL/2 (ImageNet 512×512), achieves 60% cost reduction with 16.1% FID improvement on PixArt-$\alpha$ (MSCOCO), and delivers 5× speedup on FLUX while improving CLIP Score over the 50-step baseline. Our key contributions are:

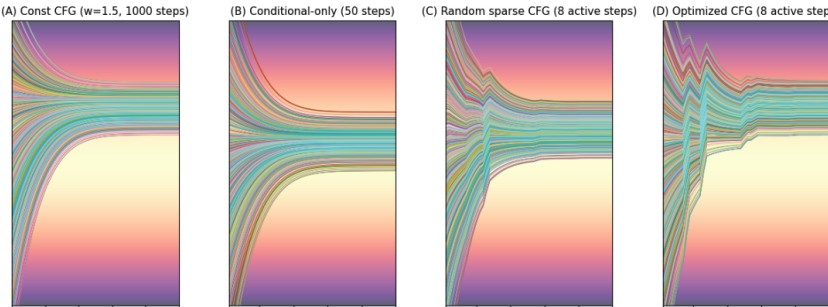

Figure 2: **1D example of OUSAC uses fewer steps to converge to the same solution as constant CFG.** (A) Constant CFG ($w$=1.5, 1000 steps) requires 2000 forward passes to map from a prior distribution (left) to a target distribution (right). (B) Conditional-only (50 steps) converges to an incorrect distribution. (C) Random sparse CFG with randomly assigned guidance scales at 8 steps also fails. (D) Our optimized sparse CFG with carefully tuned guidance scales at 8 steps matches the target distribution while using only 58 forward passes.

- The first framework to jointly optimize *which* timesteps to skip (discrete) and *what scale* to use (continuous), bridging the efficiency-focused and quality-focused directions in CFG research.
- Evolutionary optimization that searches the hybrid discrete-continuous space without backpropagation, enabling optimization on large-scale models where gradient-based methods are infeasible.
- Adaptive rank allocation via coordinate descent, the first integration of guidance scheduling with feature caching, addressing the unexplored incompatibility between variable guidance and standard caching methods.

## 2 RELATED WORK

**Diffusion model acceleration.** Recent acceleration methods fall into three categories. Sampling acceleration reduces denoising steps through improved numerical solvers. DDIM (Song et al., 2022) enables deterministic sampling with fewer steps, while DPM-Solver (Lu et al., 2022) uses exponential integrators for faster convergence. Higher-order methods (Zhang & Chen, 2023) and progressive distillation (Salimans & Ho, 2022) further reduce steps, though distillation requires expensive training. Recent inference-time distillation (Park et al., 2024) eliminates separate training but still cannot generalize across guidance scales. Architectural optimizations reduce per-step costs through efficient designs (Peebles & Xie, 2023), pruning (Zhu et al., 2024), and quantization (Liu et al., 2025c; Yang et al., 2025). Our work maintain full denoising steps while reducing per-step cost through selective CFG forward passes and adaptive caching, without architectural modifications.

**Dynamic guidance scheduling.** Evidence shows constant CFG wastes computation. Kynkäänniemi et al. (2024) find guidance harmful at extreme noise levels, while Wang et al. (2024) shows monotonic schedules outperform constant guidance. Theoretical advances include progressive guidance (Xi et al., 2024), characteristic guidance with non-linear corrections (Zheng & Lan, 2024), and gradient artifact correction (Gao et al., 2025). Methods for reducing CFG cost include convergence detection (Castillo et al., 2023), early-stage compression (Dinh et al., 2024), and adaptive scaling (Malarz et al., 2025b; Li et al., 2025). Other works explore time-varying scales for quality improvement (Gao et al.; Malarz et al., 2025a; Zhu et al., 2025; Sadat et al., 2025). Zhang et al. (2025) and Yehezkel et al. (2025) showed optimal schedules vary across architectures. Alternative approaches include autoguidance (Karras et al., 2024) and condition annealing (Sadat et al., 2023). We are the first to use evolutionary optimization to jointly search discrete skip patterns and continuous guidance scales.

**Feature caching and calibration.** Caching exploits temporal redundancy between timesteps. Deep-Cache (Ma et al., 2023) pioneered feature reuse for U-Nets. TGATE (Liu et al., 2024) reveals that cross-attention converges early while self-attention becomes crucial later, enabling selective attention caching. Extensions to transformers include block-level caching (Wimbauer et al., 2024) and training-inference harmonization (Huang et al., 2025). Δ-DiT (Chen et al., 2024) observes that

front DiT blocks handle outlines while rear blocks refine details, caching different blocks at different sampling stages accordingly. Learning-to-Cache (Ma et al., 2024) uses learned routing but produces fixed patterns. Token-wise caching (Zou et al., 2025) achieves 2.36x speedup through selective token reuse. TeaCache (Liu et al., 2025a) leverages timestep embeddings to estimate output differences for adaptive caching in video diffusion, while TaylorSeer (Liu et al., 2025b) predicts future features via Taylor expansion rather than direct reuse. ICC (Chen et al., 2025) combines caching with uniform SVD calibration across all blocks. However, no existing work addresses how calibration should adapt when guidance varies.

## 3 PRELIMINARIES

**Diffusion models and sampling.** Diffusion models learn to reverse a forward noising process defined as $q(x_t|x_0) = \mathcal{N}(x_t; \sqrt{\bar{\alpha}_t}x_0, (1 - \bar{\alpha}_t)I)$, where $x_t$ is the noised image at timestep $t \in \{1, ..., T\}$, $x_0$ is the clean image, and $\bar{\alpha}_t$ represents the cumulative noise schedule. The denoising process can be accelerated using DDIM (Song et al., 2022):

$$x_{t-1} = \sqrt{\bar{\alpha}_{t-1}}\hat{x}_0(x_t, t) + \sqrt{1 - \bar{\alpha}_{t-1} - \sigma_t^2} \cdot \epsilon_\theta(x_t, t) + \sigma_t \epsilon_t, \tag{1}$$

where $\hat{x}_0$ is the predicted clean image from $x_t$, $\epsilon_\theta$ is the learned noise predictor network, $\sigma_t$ controls the stochasticity of sampling (with $\sigma_t = 0$ for deterministic generation), and $\epsilon_t \sim \mathcal{N}(0, I)$.

**Classifier-free guidance.** To improve the quality of conditional generation, CFG (Ho & Salimans, 2022) interpolates between conditional and unconditional predictions:

$$\tilde{\epsilon}_\theta(x_t, c, t) = \epsilon_\theta(x_t, \emptyset, t) + w \cdot (\epsilon_\theta(x_t, c, t) - \epsilon_\theta(x_t, \emptyset, t)), \tag{2}$$

where $c$ denotes the conditioning information, $\emptyset$ represents null conditioning, and $w$ controls the guidance scale. By applying CFG at each timestep, the total computation is *doubled*.

**Caching for diffusion models.** Recent work (Chen et al., 2025) accelerates diffusion transformers by caching and reusing features across timesteps. The method corrects cached features through layer-wise calibration:

$$\hat{\mathbf{h}}_{\text{out}}^\ell = \mathcal{P}(\mathbf{h}_{\text{out}}^{\ell,\text{prev}}) + \mathbf{A}^\ell(\mathbf{h}_{\text{in}}^\ell - \mathcal{P}(\mathbf{h}_{\text{in}}^{\ell,\text{prev}})), \tag{3}$$

where $\ell$ is the layer index, $\mathcal{P}(\cdot)$ denotes the caching operation from the previous timestep, $\mathbf{h}_{\text{in}}^\ell$ is the current input to layer $\ell$, and $\mathcal{P}(\mathbf{h}_{\text{in}}^{\ell,\text{prev}})$ and $\mathcal{P}(\mathbf{h}_{\text{out}}^{\ell,\text{prev}})$ are the cached input and output from the previous timestep. Each layer has its own calibration matrix $\mathbf{A}^\ell$ that transforms the input increment to correct the cached output. To reduce computation, each $\mathbf{A}^\ell$ is approximated using SVD decomposition: $\mathbf{A}^\ell = \mathbf{U}^\ell \mathbf{\Sigma}^\ell \mathbf{V}^{\ell T} \approx \mathbf{U}_r^\ell \mathbf{\Sigma}_r^\ell \mathbf{V}_r^{\ell T}$, where the subscript $r$ denotes truncation to rank $r$. Prior increment-calibrated caching methods use uniform rank $r$ across all layers in all transformer blocks.

## 4 OUSAC

OUSAC accelerates diffusion transformers through two-stage optimization. *Stage 1* uses evolutionary algorithms to discover sparse guidance schedules that eliminate unconditional forward passes at non-critical timesteps where guidance contributes minimally to generation quality. *Stage 2* develops adaptive rank allocation for feature caching, where different transformer regions receive different calibration ranks to handle the varying feature differences introduced by variable guidance patterns. We optimize these components once per pre-trained model to discover optimal configurations. During inference, no optimization occurs—we simply apply the discovered sparse guidance schedule and adaptive caching configuration to accelerate generation. The discovered patterns generalize across different prompts and conditions. Sections 4.1 and 4.2 detail each optimization stage.

### 4.1 STAGE 1: DISCOVERING SPARSE GUIDANCE SCHEDULES

#### 4.1.1 PROBLEM FORMULATION

CFG's computational cost comes from applying guidance uniformly at every denoising timestep. Recent empirical studies (Kynkäänniemi et al., 2024) have provided valuable insights showing that

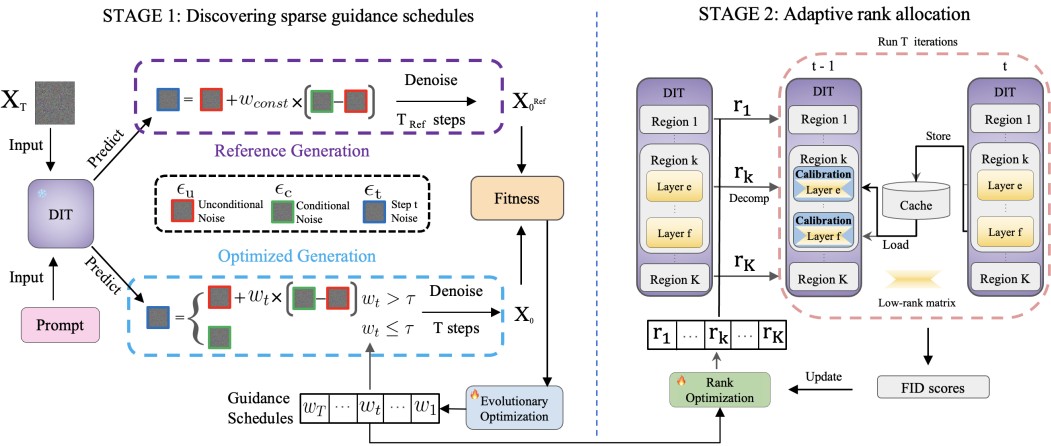

Figure 3: **The two-stage OUSAC optimization framework. Stage 1 (Left):** Evolutionary optimization discovers sparse guidance schedules by refining per-step guidance $\mathbf{w} = [w_T, \ldots, w_1]$. Starting from noise $x_T$, the framework generates a reference $x_0^{\text{Ref}}$ via $T_{\text{Ref}}$ denoising steps. At each timestep, full CFG is applied if $w_t > \tau$, otherwise only conditional forward passes are performed. The fitness function balances quality and sparsity, iteratively improving $\mathbf{w}$ through population sampling and evaluation. **Stage 2 (Right):** Adaptive rank allocation optimizes caching. DiT blocks are partitioned into $K$ regions, each assigned a calibration rank $r_k$. Cached features are corrected via SVD-based calibration. Coordinate descent with binary search tunes ranks to minimize FID.

guidance can be harmful at extreme noise levels and unnecessary near convergence. These pioneering works establish important foundations through interval-based strategies that significantly improve efficiency. In this work, we take it one step further and explore if we can find more complex, flexible, and task-specific patterns for CFG. We start by replacing the constant guidance scale $w$ with a per-timestep guidance schedule to reformulate Equation 2:

$$\tilde{\epsilon}_\theta(x_t, c, t) = \epsilon_\theta(x_t, \emptyset, t) + w_t \cdot (\epsilon_\theta(x_t, c, t) - \epsilon_\theta(x_t, \emptyset, t)), \tag{4}$$

where $w_t$ is now timestep-dependent. We optimize a guidance schedule $\mathbf{w} = [w_1, w_2, ..., w_T]$ where each $w_t \in [0, w_{\max}]$. When $w_t$ falls below a threshold $\tau$, we set $w_t = 0$ and skip the unconditional forward pass entirely, performing only the conditional forward pass.

The best schedule $\mathbf{w}$ can be found by solving the following optimization problem:

$$\mathbf{w}^* = \arg\min_{\mathbf{w}} \mathcal{L}_{\text{total}}(\mathbf{w}) = \mathcal{L}_{\text{quality}}(\mathbf{w}) + \lambda \mathcal{L}_{\text{sparse}}(\mathbf{w}) \tag{5}$$

The quality preservation term $\mathcal{L}_{\text{quality}}$ ensures our sparse schedule maintains generation fidelity through output matching:

$$\mathcal{L}_{\text{quality}}(\mathbf{w}) = \mathbb{E}_{x_T, c} \left[ \|\mathcal{G}_T(x_T, c; \mathbf{w}) - \mathcal{G}_{T_{\text{ref}}}(x_T, c; w_{\text{const}})\|_2^2 \right] \tag{6}$$

where $\mathcal{G}_T$ denotes the $T$-step generation process starting from initial noise $x_T$ with our optimized schedule $\mathbf{w}$, and $\mathcal{G}_{T_{\text{ref}}}$ represents reference generation from the same $x_T$ using constant guidance $w_{\text{const}}$. The reference uses substantially more steps ($T_{\text{ref}} \gg T$, typically 1000 vs 20-50) to provide smooth denoising process and high-quality targets. Starting from identical noise ensures fair comparison and helps identify critical timesteps for guidance.

The sparsity term directly penalizes the number of timesteps requiring full CFG forward pass: $\mathcal{L}_{\text{sparse}}(\mathbf{w}) = \sum_{t=1}^{T} \mathbb{I}[w_t > \tau]$ where $\tau$ serves as an activation threshold below which guidance is completely disabled, eliminating the unconditional forward pass. This binary decision at each timestep transforms the optimization into a hybrid continuous-discrete problem (Barton et al., 2000).

### 4.1.2 EVOLUTIONARY OPTIMIZATION STRATEGY

Direct gradient-based optimization of this objective is intractable as it would require backpropagation through the entire $T$-step generation trajectory, creating prohibitive memory requirements and

suffering from vanishing gradients. Instead, we employ a tailored evolutionary strategy that operates in a transformed space for numerical stability.

We maintain a population center $\boldsymbol{\mu} \in \mathbb{R}^T$ where $\boldsymbol{\mu} = [\mu_1, \ldots, \mu_T]^T$ with each $\mu_t \in \mathbb{R}$. This center represents the mean of our search distribution in the parameter space, a fundamental concept in both CMA-ES (Hansen, 2023) and Natural Evolution Strategies (Wierstra et al., 2014; Yi et al., 2009). At each generation $g \in \{1, \ldots, G\}$, we decode the center to get base guidance values:

$$\mathbf{w}_{\text{base}} = w_{\max} \cdot \text{sigmoid}(\boldsymbol{\mu_g}) \tag{7}$$

We construct a population by perturbing these base values. For each candidate $i \in \{1, \ldots, P\}$: $\mathbf{w}^{(i)} = \mathbf{w}_{\text{base}} + \boldsymbol{\delta}^{(i)}$, where $\boldsymbol{\delta}^{(i)} \sim \mathcal{N}(0, \sigma_{\text{noise}}^2 I)$ and $\sigma_{\text{noise}} = \sigma_0(1 - g/G)$ decreases across generations to refine exploration, with $\sigma_0$ being the initial noise scale.

We apply a threshold $\tau$ to sparsify the guidance schedule and determine the noise prediction at $t$:

$$\epsilon_t = \begin{cases} \epsilon_u + w_t^{(i)} \cdot (\epsilon_c - \epsilon_u) & \text{if } w_t^{(i)} \geq \tau \\ \epsilon_c & \text{if } w_t^{(i)} < \tau \end{cases} \tag{8}$$

Each candidate is evaluated using: $f^{(i)} = -\mathcal{L}_{\text{quality}}(\mathbf{w}^{(i)}) + \lambda \cdot S(\mathbf{w}^{(i)})$, where $S(\mathbf{w}^{(i)}) = (T - \|\mathbf{w}^{(i)}\|_0)/T$ measures sparsity. With fitness for all candidates $\{f^{(1)}, \cdots, f^{(P)}\}$, we compute rank-based weights: $a_i = d_i/(P-1) - 0.5$, where $d_i \in \{0, 1, \cdots, P-1\}$ is the rank of candidate $i$, with 0 for the lowest fitness and $P - 1$ for the highest. This rank-based weighting scheme follows established practices in evolution strategies (Hansen & Ostermeier, 2001; Hansen et al., 2003).

The population center evolves through natural gradient estimation:

$$\boldsymbol{\mu_{g+1}} \leftarrow \boldsymbol{\mu_g} + \frac{\eta}{P} \sum_{i=1}^{P} a_i \cdot (\text{sigmoid}^{-1}(\mathbf{w}^{(i)}/w_{\max}) - \boldsymbol{\mu_g}) \tag{9}$$

After $G$ generations, the converged center $\boldsymbol{\mu}^*$ yields: $\mathbf{w}^* = w_{\max} \cdot \text{sigmoid}(\boldsymbol{\mu}^*)$. This sparse schedule $\mathbf{w}^*$ applies guidance selectively at critical timesteps to reduce redundant computations.

## 4.2 STAGE 2: ADAPTIVE CACHING UNDER DENOISING FLUCTUATIONS

### 4.2.1 THE CHALLENGE OF CACHING WITH VARIABLE GUIDANCE

The sparse guidance schedules discovered in *Stage 1* fundamentally challenge existing caching methods. Standard caching approaches such as ICC (Chen et al., 2025) assume that features between consecutive timesteps remain similar. This assumption holds when guidance remains constant throughout denoising. However, our variable guidance patterns from *Stage 1* break this assumption, causing the incremental calibration in Equation 3 to become less effective when correcting the larger feature differences introduced by changing guidance scales.

As shown in Figure 5, when we apply variable guidance from *Stage 1* with naive caching, feature reconstruction errors increase significantly compared to constant CFG with the same caching method. The variable guidance causes higher MSE across all transformer blocks, with particularly severe degradation in deeper blocks. Since these elevated reconstruction errors accumulate through the network and degrade final image quality, which motivates us to explore the reason behind this.

To address this challenge, we first need to understand how variable guidance affects the denoising process. We consider two scenarios: when consecutive timesteps both use CFG but with different guidance scales, and when timesteps switch between using CFG and using only conditional prediction. Each scenario introduces distinct types of denoising fluctuations that degrade caching.

*Scenario 1: Both timesteps use CFG with different scales.* When both $w_t^* > \tau$ and $w_{t-1}^* > \tau$, the denoising process deviates by:

$$\Delta x_{t-1}^{\text{strength}} = \sqrt{\bar{\alpha}_{t-1}/\bar{\alpha}_t} \cdot (w_{t-1}^* - w_t^*) \cdot [\epsilon_c(x_t, t) - \epsilon_u(x_t, t)] \tag{10}$$

where $\bar{\alpha}_t$ is the cumulative noise schedule coefficient. This deviation grows with the guidance difference $(w_{t-1}^* - w_t^*)$ and the gap between conditional and unconditional predictions.

*Scenario 2: CFG at timestep $t$ transitions to no guidance at timestep $t - 1$.* When $t$ uses CFG ($w_t^* > \tau$) but $t - 1$ uses only conditional prediction ($w_{t-1}^* < \tau$), the deviation becomes:

$$\Delta x_{t-1}^{\text{switch}} = \beta_{t-1,t} \cdot (1 - w_t^*) \cdot [\epsilon_c(x_t, t) - \epsilon_u(x_t, t)], \tag{11}$$

where $\beta_{t-1,t} = \sqrt{1 - \bar{\alpha}_{t-1}} - \sqrt{\bar{\alpha}_{t-1}(1 - \bar{\alpha}_t)/\bar{\alpha}_t}$ is the noise coefficient for DDIM sampling. These deviations exceed what standard caching methods can handle, as cached features from timestep $t$ no longer match the expected input for timestep $t-1$. This mismatch causes higher reconstruction errors that accumulate through the transformer blocks. This heterogeneous error distribution reveals that different transformer regions require different calibration to effectively handle variable guidance.

### 4.2.2 REGION-ADAPTIVE RANK ALLOCATION

Caching with variable guidance causes larger denoising deviations than caching under a constant CFG. While incremental calibration was designed to alleviate caching errors under constant CFG, we now examine how it performs under variable guidance patterns. In our experiment, we find that different blocks benefit from different calibration ranks. Early (block 0) and late blocks (block 26) maintain lower MSE with higher ranks (=512), while middle blocks (block 12) achieve better performance with lower ranks (=256). This heterogeneous pattern shows that uniform rank allocation cannot handle the varying error magnitudes introduced by variable guidance. For detailed analysis across all transformer blocks, please refer to the appendix A.2. Since higher ranks directly increase computational cost during denoising, we face a trade-off between feature consistency and efficiency. These observations motivate our systematic approach to discovering optimal rank distributions, where we assign each region the rank that best balances error reduction and compute.

Formally, we partition the $N$ transformer blocks into $K$ regions $\mathcal{R} = \{R_1, \ldots, R_K\}$ based on their network position. We divide blocks uniformly such that each region contains $\lfloor N/K \rfloor$ consecutive blocks, with region $R_k$ containing blocks from index $(k - 1) \cdot \lfloor N/K \rfloor$ to $k \cdot \lfloor N/K \rfloor - 1$. Each region $k$ receives a tailored calibration rank $r_k$ to obtain a region-specific calibration matrix:

$$\mathbf{A}_\ell \approx \mathbf{U}_{\ell,r_k} \mathbf{\Sigma}_{\ell,r_k} \mathbf{V}_{\ell,r_k}^T \quad \text{for layer } \ell \in R_k \tag{12}$$

We optimize the rank configuration $\mathbf{r} = [r_1, r_2, \ldots, r_K]$ where each $r_k \in [r_{\min}, r_{\max}]$.

### 4.2.3 RANK OPTIMIZATION VIA COORDINATE DESCENT.

Finding the optimal rank configuration $\mathbf{r}^* = [r_1, \ldots, r_K]$ requires searching over a large discrete space where each region can take ranks from $[r_{\min}, r_{\max}]$. This is challenging because rank assignments across regions interact through the sequential nature of the transformer: early blocks affect later blocks' inputs, creating complex dependencies that make the relationship between rank configuration and final generation quality non-linear. This is a constrained optimization problem:

$$\mathbf{r}^* = \arg\min_{\mathbf{r}} \text{FID}(\mathcal{G}_T(\mathbf{r}, \mathbf{w}^*)) \quad \text{s.t.} \quad r_k \in [r_{\min}, r_{\max}], \quad \sum_{k=1}^K r_k \leq B, \tag{13}$$

where $\mathcal{G}_T(\mathbf{r}, \mathbf{w}^*)$ denotes the $T$-step generation process using rank configuration $\mathbf{r}$ with the optimized guidance schedule $\mathbf{w}^*$ from *Stage 1*, and $B$ represents the total computational budget. The generation process follows the same $T$-step denoising trajectory as in *Stage 1*, but now using regional calibration matrices based on the rank configuration $\mathbf{r}$.

We solve this optimization through coordinate descent (Wright, 2015), which naturally decomposes the problem into a sequence of single-variable optimizations. For each region $k$, we fix the ranks of all other regions and search for the optimal $r_k$ that minimizes the FID score:

$$r_k^* = \arg\min_{r_k} \text{FID}(\mathcal{G}(r_1, \ldots, r_k, \ldots, r_K, \mathbf{w}^*)) \tag{14}$$

Within each coordinate optimization step, we use binary search to efficiently explore the rank space $[r_{\min}, r_{\max}]$. This procedure iterates across all regions until the overall rank configuration converges.

## 5 EXPERIMENTS

### 5.1 EXPERIMENTAL SETUP

**Models and datasets.** We evaluate OUSAC on three state-of-the-art diffusion transformers. For class-conditional generation, we employ DiT-XL/2 (Peebles & Xie, 2023) on ImageNet (Deng et al.,

Table 1: Quantitative comparison of acceleration methods on DiT-XL/2 for ImageNet generation.

| Methods | Steps | CFG-Steps↓ | MACs (T)↓ | Latency (s)↓ | IS↑ | FID↓ | sFID↓ | Prec.↑ | Recall↑ |
|---|---|---|---|---|---|---|---|---|---|
| **DiT-XL/2 (ImageNet 512×512)** | | | | | | | | | |
| DDIM | 1000 | 1000 | 1049.1 | 416.79 | 210.6 | 2.99 | 4.38 | 83.17 | 55.6 |
| DDIM | 50 | 50 | 52.45 | 20.86 | 203.8 | 3.20 | 4.53 | 83.27 | 56.4 |
| L2C | 50 | 50 | 40.62 | 16.30 | 199.5 | 3.98 | 5.66 | 82.46 | 53.3 |
| ICC | 50 | 50 | 33.43 | 15.88 | 200.0 | 3.73 | 5.39 | 83.30 | 55.6 |
| TaylorSeer ($N=3, O=3$) | 50 | 50 | 18.88 | 12.08 | 201.2 | 3.51 | 4.37 | 83.46 | 53.3 |
| OUSAC w/o cache | 50 | 9 | 30.94 | 12.91 | 228.3 | **2.71** | 4.38 | 83.43 | **57.0** |
| OUSAC | 50 | 9 | **24.93** | **11.28** | 228.6 | 2.72 | **4.13** | 83.62 | 55.3 |
| DDIM | 30 | 30 | 31.47 | 12.52 | 198.0 | 3.86 | 4.94 | **83.12** | 54.4 |
| L2C | 30 | 30 | 25.72 | 10.31 | 189.4 | 4.93 | 6.72 | 82.14 | 54.5 |
| ICC | 30 | 30 | 20.05 | 9.54 | 171.0 | 6.85 | 6.72 | 79.84 | 53.5 |
| OUSAC w/o cache | 30 | 8 | 19.93 | 8.20 | **214.8** | 3.33 | 4.85 | 82.99 | 55.3 |
| OUSAC | 30 | 8 | **16.01** | **7.19** | 209.7 | 3.37 | **4.66** | 82.62 | **56.5** |
| **DiT-XL/2 (ImageNet 256×256)** | | | | | | | | | |
| DDIM | 1000 | 1000 | 237.2 | 106.42 | 245.0 | 2.12 | 4.66 | 80.66 | 59.7 |
| DDIM | 50 | 50 | 11.86 | 5.33 | 239.4 | 2.23 | 4.29 | 80.06 | 59.2 |
| HarmoniCa | 50 | 50 | 10.58 | 4.78 | 210.1 | 3.33 | 5.03 | 77.40 | **60.1** |
| L2C | 50 | 50 | 9.76 | 4.43 | 245.5 | 2.23 | **4.27** | 80.95 | 59.1 |
| ICC | 50 | 50 | 8.05 | 4.18 | 258.5 | 2.16 | 4.28 | 82.08 | 58.1 |
| OUSAC w/o cache | 50 | 8 | 6.63 | 3.37 | 263.0 | 2.10 | 4.29 | 81.85 | 58.8 |
| OUSAC | 50 | 8 | **5.07** | **2.81** | 266.5 | **2.04** | 4.29 | **82.09** | 58.7 |

2009), generating 50,000 images at both $256\times256$ and $512\times512$ resolutions across all 1,000 classes. For text-to-image synthesis, we utilize PixArt-$\alpha$ (Chen et al., 2023) on MSCOCO 2014 (Lin et al., 2015), producing 30,000 images at $256\times256$ resolution using the caption set from (Zou et al., 2025), and FLUX (Labs, 2024) with True CFG on DrawBench (Saharia et al., 2022) and GenEval (Ghosh et al., 2023) at $512 \times 512$ resolution.

**Evaluation metrics.** For ImageNet generation, we adopt the standard evaluation suite: Inception Score (IS) (Salimans et al., 2016) for sample quality, Fréchet Inception Distance (FID) (Nash et al., 2021) and spatial FID (sFID) for distribution matching, and Precision-Recall (Kynkäänniemi et al., 2019) for mode coverage and fidelity. For MSCOCO text-to-image synthesis, we report FID-30k and CLIP Score (Hessel et al., 2022) to assess both visual quality and text-image alignment, along with CLIP Score on PartiPrompts (Yu et al., 2022) for complex prompt adherence. For FLUX (Labs, 2024), we report ImageReward (Xu et al., 2023) for human preference alignment and CLIP Score for text-image consistency on DrawBench (Saharia et al., 2022), and compositional accuracy metrics on GenEval (Ghosh et al., 2023). Computational efficiency is quantified through multiply-accumulate operations (MACs), providing hardware-agnostic performance measurements. Latency is measured with batch size 8 on a single H100 GPU.

**Baselines.** We compare against representative acceleration techniques: (1) ICC (Chen et al., 2025), which applies uniform-rank increment-calibrated caching to transformer blocks; (2) Learning-to-Cache (L2C) (Ma et al., 2024), which uses learned routing for adaptive feature caching; (3) HarmoniCa (Huang et al., 2025), which harmonizes training and inference through optimized caching strategies; (4) TaylorSeer (Liu et al., 2025b), which predicts future features via Taylor expansion rather than direct reuse; and (5) standard DDIM sampling at various step counts as reference points. These baselines represent both training-free methods (ICC, TaylorSeer) and approaches requiring additional training (L2C, HarmoniCa), enabling comprehensive evaluation of our gradient-free optimization approach. Since parts of L2C and HarmoniCa lack publicly available checkpoints for our experimental settings, we reimplemented them following their published protocols.

Table 2: Performance evaluation on MSCOCO 2014 with PixArt-$\alpha$ at $256 \times 256$ resolution.

| Method | Steps | CFG-Steps↓ | MACs (T)↓ | FID↓ | CLIP$_{COCO}$↑ | CLIP$_{Parti}$↑ |
|---|---|---|---|---|---|---|
| DPM-Solver | 1000 | 1000 | 336.11 | 22.97 | 16.42 | 17.31 |
| DPM-Solver | 20 | 20 | 6.72 | 24.60 | 16.31 | 17.36 |
| ICC | 20 | 20 | 3.70 | 21.86 | 16.47 | 17.20 |
| OUSAC w/o cache | 20 | **6** | 4.37 | 22.69 | 16.39 | **17.92** |
| OUSAC | 20 | **6** | **2.67** | **19.27** | **16.48** | 17.45 |

Table 3: Performance evaluation on DrawBench with FLUX at $512 \times 512$ resolution.

| Method | Steps | CFG-Steps↓ | Latency(s)↓ | MACs(T)↓ | ImageReward↑ | CLIP Score↑ |
|---|---|---|---|---|---|---|
| FLUX (True CFG=1.5) | 50 | 50 | 52.01 | 1143.82 | 0.9956 | 27.70 |
| FLUX (True CFG=1.5) | 20 | 20 | 21.24 | 457.52 | 0.8950 | 27.55 |
| FLUX (True CFG=1.5) | 16 | 16 | 17.19 | 366.02 | 0.7866 | 27.46 |
| FLUX | 50 | × | 26.43 | 571.48 | 0.9296 | 27.23 |
| FLUX | 20 | × | 10.87 | 228.76 | 0.8934 | 27.03 |
| TaylorSeer ($N=3, O=2$) | 50 | 50 | 28.30 | 411.77 | 1.0022 | 27.78 |
| OUSAC w/o cache | 20 | 8 | 15.15 | 320.26 | **1.0092** | **28.06** |
| TaylorSeer ($N=6, O=2$) | 50 | 50 | 20.10 | 228.76 | 0.8492 | 27.26 |
| TaylorSeer ($N=3, O=2$) | 50 | × | 13.87 | 205.88 | 0.9445 | 27.37 |
| OUSAC | 20 | 8 | 14.88 | 216.48 | **0.9726** | **28.06** |

## 5.2 MAIN RESULTS

**DiT-XL/2 on ImageNet.** Table 1 demonstrates substantial efficiency gains across both resolutions. At $512 \times 512$, OUSAC matches 50-step DDIM quality (FID 2.72 vs 3.20) with 47% less computation (24.97T vs 52.45T MACs) by applying guidance at only 9 of 50 timesteps. It even surpasses 1000-step DDIM (FID 2.72 vs 2.99) while using 97% less computation. At $256 \times 256$, OUSAC achieves the best FID (2.04) among all baselines, including 1000-step DDIM (2.12), while using only 5.07T vs 11.86T MACs for standard 50-step sampling.

**PixArt-$\alpha$ on MSCOCO.** Text-to-image generation with DPM-Solver shows a minimal quality gap between 20- and 1000-step sampling (FID 24.60 vs 22.97), providing weak learning signals for our evolutionary optimization. Despite this challenge, OUSAC discovers that only 6 out of 20 timesteps need guidance on average, achieving FID 19.27 (21.7% improvement) with 60% computational reduction (2.67 vs 6.72 MACs), while maintaining text-image alignment (CLIP score 16.48 vs 16.31).

**FLUX on DrawBench and GenEval.** Tables 3 and 4 evaluate OUSAC on FLUX with True CFG. OUSAC discovers that only 8 out of 20 timesteps require guidance. On DrawBench, OUSAC without caching achieves the highest ImageReward (1.0092) and CLIP Score (28.06), outperforming 50-step FLUX with full CFG (0.9956, 27.70) while using 72% less computation (320T vs 1144T MACs). With adaptive caching, OUSAC further reduces latency to 14.88s while maintaining competitive quality. On GenEval, OUSAC achieves an overall score of 67.46 without caching, approaching the 68.60 of 50-step full CFG, and notably excels on Color-Attr (51.50 vs 49.75). These results confirm that OUSAC's sparse guidance patterns transfer effectively to state-of-the-art diffusion transformers.

**Comparison with other CFG redundancy reduction methods.** Table 10 compares OUSAC with Adaptive Guidance (Castillo et al., 2023), which uses discrete selection among $k+2$ predetermined options per timestep. While Adaptive Guidance achieves 18% computational reduction (32 CFG steps), OUSAC's continuous optimization ($w_t \in [0, w_{max}]$) discovers only 9 timesteps need guidance, achieving 41% reduction and improving FID from 3.25 to 2.71.

## 5.3 ABLATION STUDY

We systematically investigate the design choices in OUSAC through controlled experiments, analyzing how each component contributes to the overall performance gains.

Table 4: Performance evaluation on GenEval with FLUX at $512 \times 512$ resolution.

| Method | Steps | CFG-Steps↓ | Latency(s)↓ | Position | Colors | Counting | Color-Attr | Two-Obj | Single-Obj | Overall↑ |
|---|---|---|---|---|---|---|---|---|---|---|
| FLUX (True CFG=1.5) | 50 | 50 | 52.01 | 19.50 | **80.32** | **77.19** | 49.75 | **86.11** | 98.75 | **68.60** |
| FLUX (True CFG=1.5) | 20 | 20 | 21.24 | **22.50** | 74.47 | 74.69 | 42.50 | 84.34 | 97.81 | 66.05 |
| FLUX (True CFG=1.5) | 16 | 16 | 17.19 | 22.00 | 73.40 | 68.75 | 35.50 | 84.90 | 97.81 | 63.59 |
| FLUX | 50 | ✗ | 26.43 | 17.25 | 77.39 | 73.44 | 45.75 | 80.05 | 98.44 | 65.38 |
| FLUX | 20 | ✗ | 10.87 | 20.75 | 78.99 | 69.69 | 44.25 | 79.80 | 98.12 | 65.26 |
| TaylorSeer ($N = 3$, $O = 2$) | 50 | 50 | 28.30 | 17.75 | 78.99 | 74.69 | 50.50 | 85.86 | 98.44 | 67.70 |
| OUSAC w/o cache | 20 | 8 | 15.15 | 21.25 | **80.32** | 68.44 | **51.50** | 84.85 | 98.44 | 67.46 |
| TaylorSeer ($N = 6$, $O = 2$) | 50 | 50 | 20.10 | 19.25 | 63.56 | 69.06 | 30.00 | 77.53 | 98.12 | 59.58 |
| TaylorSeer ($N = 3$, $O = 2$) | 50 | ✗ | 13.87 | 17.50 | 77.13 | **70.62** | **50.25** | 81.06 | **99.06** | 65.93 |
| OUSAC | 20 | 8 | 14.88 | **20.25** | 79.79 | 70.31 | 49.50 | **82.07** | 98.75 | **66.77** |

Table 5: Comparison of CFG redundancy reduction methods on DiT-XL/2 (ImageNet 512×512).

| Methods | Strategy | CFG Steps↓ | MACs (T)↓ | ΔMACs↓ | IS↑ | FID↓ | sFID↓ | Precision↑ | Recall↑ |
|---|---|---|---|---|---|---|---|---|---|
| DDIM | Constant | 50 | 52.45 | – | 203.8 | 3.25 | 4.53 | 83.27 | 56.4 |
| Adaptive Guidance | Discrete | 32 | 43.00 | -18% | 179.2 | 4.15 | 4.68 | 82.88 | 55.9 |
| OUSAC | Continuous | **9** | **30.94** | **-41%** | **228.3** | **2.71** | **4.38** | **83.43** | **57.0** |

Table 6: Ablation study of rank allocation strategies for DiT-XL/2 (ImageNet 512×512).

| Methods | Steps | MACs (T)↓ | IS↑ | FID↓ | sFID↓ | Precision↑ | Recall↑ |
|---|---|---|---|---|---|---|---|
| DDIM | 50 | 52.45 | 203.8 | 3.25 | 4.53 | 83.27 | 56.4 |
| ICC | 50 | 33.43 | 200.0 | 3.73 | 5.39 | 83.30 | 55.6 |
| OUSAC w/o cache | 50 | 30.94 | 228.3 | **2.71** | **4.38** | 83.43 | **57.0** |
| OUSAC w/ uniform $r$=1024 | 50 | 34.68 | **229.8** | 2.92 | 4.96 | 82.12 | 54.6 |
| OUSAC w/ uniform $r$=512 | 50 | 26.16 | 205.0 | 4.46 | 5.19 | 78.19 | 55.4 |
| OUSAC w/ uniform $r$=256 | 50 | 23.94 | 213.7 | 3.68 | 6.47 | 82.61 | 54.8 |
| **OUSAC** | 50 | **22.37** | 228.1 | 3.01 | 4.63 | **83.82** | 54.9 |

**Does adaptive rank allocation outperform uniform rank allocation?** Table 6 validates our adaptive rank allocation strategy. When replacing OUSAC's adaptive ranks with uniform ranks across all blocks, performance degrades significantly. Uniform $r$=256 increases FID to 3.68, while uniform $r$=1024 achieves FID 2.73 but requires 55% more computation (34.68T vs 22.37T MACs). OUSAC discovers region-specific rank distributions that achieve FID 3.01 with only 22.37T MACs, demonstrating that different transformer regions require different calibration levels under variable guidance patterns. Additional ablation studies can be found in Appendix A.4, including reference generation length analysis, adaptive versus uniform rank allocation on DiT-XL/2 and PixArt-$\alpha$, guidance schedule transferability under varying CFG scales, INT8 quantization compatibility, optimization cost analysis, and convergence behavior of evolutionary search.

# 6 CONCLUSION

This paper demonstrates that variable guidance scales enable sparse computation in diffusion models. Unlike traditional CFG that applies a fixed scale uniformly, OUSAC jointly optimizes *when to skip* CFG (discrete) and *what scale to use* (continuous) through evolutionary search. This hybrid optimization discovers that adjusting scales at certain timesteps can compensate for skipping CFG at others, enabling both fewer total sampling steps and fewer CFG steps while maintaining quality. To address feature reconstruction errors from variable guidance, we propose adaptive rank allocation, the first integration of guidance scheduling with feature caching. The discovered schedules generalize across CFG strengths via multiplicative scaling ($k \cdot \mathbf{w}^*$). Experiments on DiT-XL/2, PixArt-$\alpha$, and FLUX confirm 50–70% computational savings while maintaining or improving generation quality.

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

# A APPENDIX

## A.1 THE USE OF LARGE LANGUAGE MODELS

We used large language models as writing assistants to improve the clarity and grammatical correctness of our manuscript. Specifically, we used LLM to refine sentence structure, correct grammatical errors, and enhance readability for audiences.

## A.2 MOTIVATION

### A.2.1 DENOISING DEVIATIONS UNDER VARIABLE GUIDANCE - MATH

The sparse guidance schedules from Stage 1 introduce denoising deviations that affect caching effectiveness. We analyze two primary scenarios that arise from our variable guidance patterns.

**Scenario 1: Guidance Scale Variation** When consecutive timesteps both apply CFG but with different scales ($w_t^* > \tau$ and $w_{t-1}^* > \tau$), we derive the resulting deviation.

Starting from the DDIM update (Song et al., 2022) with deterministic sampling ($\sigma_t = 0$):

$$x_{t-1} = \sqrt{\frac{\bar{\alpha}_{t-1}}{\bar{\alpha}_t}} x_t + \beta_{t-1,t} \cdot \tilde{\epsilon}_\theta(x_t, t) \tag{15}$$

where $\beta_{t-1,t} = \sqrt{1 - \bar{\alpha}_{t-1}} - \sqrt{\frac{\bar{\alpha}_{t-1}(1-\bar{\alpha}_t)}{\bar{\alpha}_t}}$.

With CFG, the noise prediction is:

$$\tilde{\epsilon}_\theta(x_t, t) = \epsilon_u(x_t, t) + w_t^*[\epsilon_c(x_t, t) - \epsilon_u(x_t, t)] \tag{16}$$

When guidance scale changes from $w_t^*$ to $w_{t-1}^*$, the deviation becomes:

$$\Delta x_{t-1}^{\text{scale}} = \beta_{t-1,t}(w_{t-1}^* - w_t^*)[\epsilon_c(x_t, t) - \epsilon_u(x_t, t)] \tag{17}$$

For the approximation in Equation 10 of the main text, we use $\beta_{t-1,t} \approx \sqrt{\bar{\alpha}_{t-1}/\bar{\alpha}_t}$ for clarity.

**Scenario 2: Guidance Mode Switching** When timestep $t$ uses CFG ($w_t^* > \tau$) but timestep $t-1$ switches to conditional-only ($w_{t-1}^* < \tau$), this is equivalent to switching from $w_t^*$ to $w_{t-1}^* = 1$ (since conditional-only means $\tilde{\epsilon} = \epsilon_c$).

Following the same framework, the switching deviation is:

$$\Delta x_{t-1}^{\text{switch}} = \beta_{t-1,t}(1 - w_t^*)[\epsilon_c(x_t, t) - \epsilon_u(x_t, t)] \tag{18}$$

Note that when $w_t^* > 1$, this deviation has opposite sign compared to Scenario 1, creating an abrupt trajectory change.

**Impact on Feature Caching** These deviations directly affect cached feature validity. Equations 17 and 18 reveal that variable guidance creates trajectory discontinuities that exceed uniform calibration's correction capacity, motivating our adaptive rank allocation in Stage 2.

### A.2.2 DENOISING DEVIATIONS UNDER VARIABLE GUIDANCE - VISUALIZATION

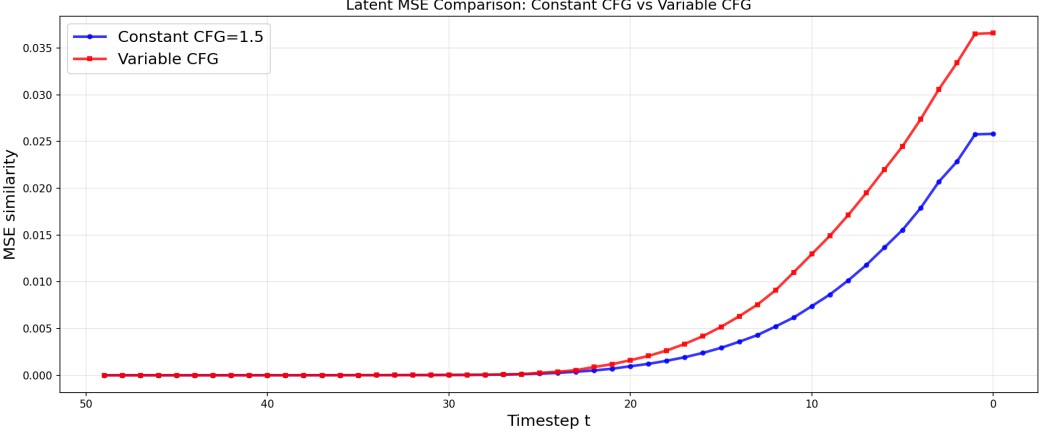

Figure 4: **Caching introduces larger errors under variable guidance.** Mean squared error between cached and non-cached outputs in the final latent space $x_0$, measured at each timestep $t$ during 50-step DDIM sampling. The MSE is computed as $\|x_0^{\text{cached}}(t) - x_0^{\text{non-cached}}(t)\|^2$. Under constant CFG ($w = 1.5$, blue line), caching maintains controlled error levels throughout the denoising process. Under variable guidance from Stage 1 (red line), caching causes accelerated error accumulation, resulting in 40% higher MSE in the final output. This demonstrates that the block-wise reconstruction errors from Figure 5 propagate through the network and accumulate into substantial quality degradation in the final latent, motivating our adaptive rank allocation strategy in Stage 2.

To empirically validate the theoretical deviations derived above, we conducted experiments measuring feature reconstruction errors under different guidance patterns. Figure 5 reveals that our sparse guidance schedule from Stage 1 introduces substantially higher reconstruction errors compared to constant CFG across all transformer blocks, with particularly severe degradation in blocks 18-26. These block-wise errors are not isolated—Figure 4 demonstrates their cumulative impact, showing that variable guidance causes 40% higher final latent error compared to constant CFG, confirming that local reconstruction errors propagate through the network to degrade generation quality. While incremental calibration can mitigate these errors, Figure 6 exposes a critical limitation of uniform calibration: different transformer regions require different calibration ranks under variable guidance. Early and late blocks achieve better performance with lower ranks ($r = 256$), while

middle blocks require higher ranks ($r = 512$) to minimize MSE. This heterogeneous error pattern across the network directly motivates our adaptive rank allocation strategy in Stage 2, which assigns region-specific calibration ranks rather than applying uniform calibration across all blocks.

### A.3 IMPLEMENTATION DETAILS

**Implementation details.** For DiT-XL/2, we use DDIM sampling with 30–50 timesteps and CFG scale 1.5. Stage-1 evolutionary optimization runs with population $P = 16$ for $G = 10$ generations at 256×256, using 32 calibration prompts and $T_{\text{ref}} = 1000$. Stage-2 partitions transformer blocks into $K = 4$ regions, searching ranks in $[16, 1024]$ with 10,000 calibration images. PixArt-$\alpha$ employs DPM-Solver with 20 steps. Stage-1 uses $P = 32$, $G = 15$ at 128×128 with 40 prompts and $T_{\text{ref}} = 1000$; Stage-2 searches $[16, 256]$ with 5,000 images. FLUX applies True CFG at scale 1.5. Stage-1 uses $P = 32$, $G = 15$ at 128×128 with 16 prompts and $T_{\text{ref}} = 100$; Stage-2 searches $[16, 512]$ with 5,000 images.

**Guidance-driven caching protocol.** We adapt the caching strategy to handle variable guidance patterns from *Stage 1*. The key modification is that when timestep $t$ uses only the conditional forward pass and timestep $t - 1$ requires full CFG, we cannot reuse cached features since the unconditional forward pass was never computed at timestep $t$. Otherwise, standard caching applies. This ensures correct CFG application while maximizing cache reuse when guidance is inactive.

### A.4 ADDITIONAL RESULTS

#### A.4.1 ABLATION STUDIES

Table 7: Effect of reference generation length on optimized guidance schedules for DiT-XL/2.

| Methods | Steps | MACs(T) | IS↑ | FID↓ | sFID↓ | Precision↑ | Recall↑ |
|---|---|---|---|---|---|---|---|
| DDIM | 1000 | 1049.1 | 210.6 | 2.99 | 4.38 | 83.17 | 55.6 |
| DDIM | 50 | 52.45 | 203.8 | 3.25 | 4.53 | 83.27 | 56.4 |
| OUSAC w/o cache ($T_{ref} = 50$) | 50 | 33.56 | 229.6 | 2.84 | 4.40 | 83.80 | 56.0 |
| OUSAC w/o cache ($T_{ref} = 500$) | 50 | 34.09 | 225.0 | 2.77 | 4.39 | 83.49 | 56.2 |
| OUSAC w/o cache ($T_{ref} = 1000$) | 50 | 30.94 | 228.3 | 2.71 | 4.38 | 83.43 | 57.0 |

**Reference generation length.** Table 7 shows that short reference generations ($T_{ref} = 50$) lead to denser schedules (33.56T MACs), while longer references ($T_{ref} = 1000$) yield sparser schedules (30.94T MACs) with better FID (2.71). Longer references help identify which timesteps truly require guidance.

Table 8: Ablation study of adaptive rank allocation strategies for DiT-XL/2.

| Methods | Steps | MACs(T) | IS↑ | FID↓ | sFID↓ | Precision↑ | Recall↑ |
|---|---|---|---|---|---|---|---|
| DiT-XL/2 (ImageNet 256×256) | | | | | | | |
| DDIM | 50 | 11.86 | 239.4 | 2.23 | 4.29 | 80.06 | 59.27 |
| ICC | 50 | 8.05 | 258.5 | 2.16 | 4.28 | 82.08 | 58.1 |
| OUSAC w/o cache | 50 | 6.63 | 263.0 | 2.10 | 4.29 | 81.85 | 58.87 |
| OUSAC ($r = 768$) | 50 | 6.88 | 273.5 | 2.12 | 4.28 | 82.89 | 57.79 |
| OUSAC ($r = 512$) | 50 | 5.81 | 276.9 | 2.48 | 4.79 | 83.17 | 56.36 |
| OUSAC ($r = 256$) | 50 | 4.74 | 271.5 | 2.27 | 4.53 | 82.97 | 57.06 |
| OUSAC | 50 | 5.07 | 266.5 | 2.04 | 4.29 | 82.09 | 58.75 |

Table 9: Ablation study of adaptive rank allocation strategies for PixArt-$\alpha$.

| Method | MACs | FID↓ | CLIP Score↑ |
|---|---|---|---|
| DPM-Solver (1000 steps) | 336.11 | 22.97 | 16.42 |
| DPM-Solver (20 steps) | 6.72 | 24.60 | 16.31 |
| OUSAC w/o cache | 4.37 | 22.69 | 16.39 |
| OUSAC ($r = 32$) | 2.63 | 20.05 | 16.49 |
| OUSAC ($r = 64$) | 2.70 | 19.92 | 16.52 |
| OUSAC | 2.67 | 19.27 | 16.48 |

Table 10: Comparison of constant CFG versus optimized sparse guidance on DiT-XL/2 (ImageNet $256 \times 256$). ($\times k$) denotes multiplying guidance scales by factor $k$: for DDIM, $w = 1.5k$; for OUSAC, the optimized schedule $\mathbf{w}^*$ is scaled element-wise as $k \cdot \mathbf{w}^*$.

| Method | CFG-Steps↓ | FID↓ | sFID↓ | Prec.↑ | Recall↑ |
|---|---|---|---|---|---|
| DDIM ($\times 1$) | 50 | 2.23 | 4.29 | 80.06 | 59.20 |
| DDIM ($\times 3.3$) | 50 | 16.39 | 15.53 | 92.22 | 22.76 |
| DDIM ($\times 5$) | 50 | 19.71 | 19.95 | 90.29 | 18.04 |
| OUSAC w/o cache ($\times 1$) | 8 | 2.10 | 4.29 | 81.85 | 58.80 |
| OUSAC w/o cache ($\times 3.3$) | 8 | 9.20 | 9.23 | 90.31 | 39.60 |
| OUSAC w/o cache ($\times 5$) | 8 | 11.76 | 13.20 | 89.53 | 33.10 |

**Adaptive versus uniform rank allocation.** Tables 8 and 9 validate our adaptive rank allocation. For DiT-XL/2 at 256×256, uniform $r = 256$ increases FID from 2.04 to 2.27, while $r = 768$ achieves 2.12 but requires 36% more computation (6.88T vs 5.07T MACs). Our coordinate descent discovers region-specific ranks achieving the best FID (2.04) with only 5.07T MACs. For PixArt-$\alpha$, adaptive allocation (FID 19.27) outperforms uniform $r = 32$ (20.05) and $r = 64$ (19.92) at comparable cost. These results confirm that different transformer regions require different calibration levels under variable guidance.

### A.4.2 ROBUSTNESS AND COMPATIBILITY

Table 11: Performance evaluation on GenEval with FLUX at $512 \times 512$ resolution. ($\times k$) denotes scaling the guidance by factor $k$: for constant CFG, $w = 1.5k$; for OUSAC, the optimized schedule $\mathbf{w}^*$ is scaled element-wise as $k \cdot \mathbf{w}^*$.

| Method | Steps | CFG-Steps↓ | Position | Colors | Counting | Color-Attr | Two-Obj | Single-Obj | Overall |
|---|---|---|---|---|---|---|---|---|---|
| FLUX w/ True CFG ($\times 1$) | 50 | 50 | 19.50 | 80.32 | 77.19 | 49.75 | 86.11 | 98.75 | 68.60 |
| FLUX w/ True CFG ($\times 2$) | 50 | 50 | 23.75 | 73.67 | 75.31 | 40.25 | 82.32 | 95.94 | 65.20 |
| FLUX w/ True CFG ($\times 3$) | 50 | 50 | 20.75 | 53.72 | 62.81 | 21.25 | 72.22 | 90.31 | 53.51 |
| OUSAC w/o cache ($\times 1$) | 20 | 8 | 21.25 | 80.32 | 68.44 | 51.50 | 84.85 | 98.44 | 67.46 |
| OUSAC w/o cache ($\times 2$) | 20 | 8 | 20.50 | 79.52 | 65.31 | 36.75 | 79.04 | 98.75 | 63.31 |
| OUSAC w/o cache ($\times 3$) | 20 | 8 | 16.50 | 72.34 | 56.88 | 23.25 | 61.87 | 96.25 | 54.51 |

**Guidance schedule transferability.** Tables 10 and 11 compare constant CFG versus OUSAC under varying guidance multipliers. On DiT-XL/2, scaling by $3.3\times$ causes severe degradation for constant CFG (FID $2.23 \to 16.39$) while OUSAC degrades gracefully ($2.10 \to 9.20$). At $5\times$ scaling, DDIM reaches FID 19.71 while OUSAC maintains 11.76. Similar robustness appears on FLUX: at $3\times$ scaling, OUSAC (54.51) slightly outperforms constant CFG (53.51). The critical timesteps identified by evolutionary optimization remain important across guidance strengths.

Table 12: Quantization compatibility on DiT-XL/2 (ImageNet 256×256) with INT8.

| Precision | Method | Steps | CFG-Steps↓ | IS↑ | FID↓ | sFID↓ |
|---|---|---|---|---|---|---|
| FP16 | DDIM | 50 | 50 | 239.4 | 2.23 | 4.29 |
| | OUSAC w/o cache | 50 | 8 | 263.0 | 2.10 | 4.29 |
| INT8 | DDIM | 50 | 50 | 199.9 | 4.61 | 9.17 |
| | OUSAC w/o cache | 50 | 8 | **225.4** | **3.61** | **8.55** |

**Quantization compatibility.** Table 12 shows OUSAC is compatible with INT8 quantization. Under INT8, OUSAC achieves FID 3.61 versus 4.61 for DDIM, and sFID 8.55 versus 9.17. This confirms that sparse guidance is orthogonal to quantization, enabling combined acceleration.

Table 13: Stage-1 evolutionary search cost analysis. All experiments use 4×H100 GPUs with parallel candidate evaluation via multiprocessing.

| | DiT-XL/2 | PixArt-$\alpha$ | FLUX |
|---|---|---|---|
| Search resolution | 256×256 | 128×128 | 128×128 |
| Population size ($P$) | 16 | 32 | 32 |
| Generations ($G$) | 10 | 15 | 15 |
| Denoising steps ($T$) | 50 | 20 | 20 |
| Reference steps ($T_{\text{ref}}$) | 1000 | 1000 | 100 |
| Calibration prompts | 32 | 40 | 16 |
| Total evaluations | 160 | 480 | 480 |
| Wall-clock time (hrs) | 4.2 | 1.1 | 1.3 |
| GPU hours (4×H100) | 16.8 | 4.4 | 5.3 |

Table 14: Stage-2 adaptive rank allocation cost analysis. All experiments use 4×H100 GPUs. Optimization is performed using the optimal guidance schedules discovered in Stage-1.

| | DiT-XL/2 | PixArt-$\alpha$ | FLUX |
|---|---|---|---|
| Search resolution | 256×256 | 256×256 | 128×128 |
| Number of regions ($K$) | 4 | 4 | 4 |
| Rank range $[r_{\min}, r_{\max}]$ | [16, 512] | [16, 256] | [16, 512] |
| Denoising steps ($T$) | 50 | 20 | 20 |
| Calibration images | 10,000 | 5,000 | 5,000 |
| Total FID evaluations | 145 | 64 | 65 |
| Wall-clock time (hrs) | 6.8 | 5.8 | 5.5 |
| GPU hours (4×H100) | 27.2 | 23.2 | 22.0 |

### A.4.3 OPTIMIZATION ANALYSIS

**Optimization cost.** Tables 13 and 14 report one-time optimization costs. Stage-1 evolutionary search takes 4–17 GPU hours depending on the model, while Stage-2 rank allocation requires 22–27 GPU hours. The total cost (<45 GPU hours per model) is amortized over all inference runs, as configurations generalize across prompts without re-optimization.

**Convergence and hyperparameters.** Figure 7 shows evolutionary optimization converges rapidly, with fitness plateauing around generation 20 and MSE stabilizing at 0.16–0.18. We use 15 generations to balance quality and cost. Figure 8 shows that while $K = 7$ achieves the lowest FID (2.03), $K = 4$ offers comparable quality (2.05) with fewer evaluations, which we adopt across experiments.

## A.5 ADDITIONAL QUALITATIVE RESULTS

Figure 9 provides visual comparisons across 15 ImageNet classes on DiT-XL/2, demonstrating that OUSAC achieves comparable visual quality to standard DDIM while using only 35% of the computational budget.

We further evaluate OUSAC on text-to-image models with both short and long prompts. Figure 10 and Figure 11 show qualitative comparisons on FLUX and PixArt-$\alpha$ respectively, where OUSAC maintains visual fidelity comparable to baselines across diverse prompt complexities.

To demonstrate the generalization of our optimized guidance schedules, Figure 12 and Figure 13 show results when scaling the CFG strength beyond the training configuration. OUSAC maintains stable generation quality across different CFG scales, indicating that the discovered sparse guidance patterns transfer well to varying guidance intensities.

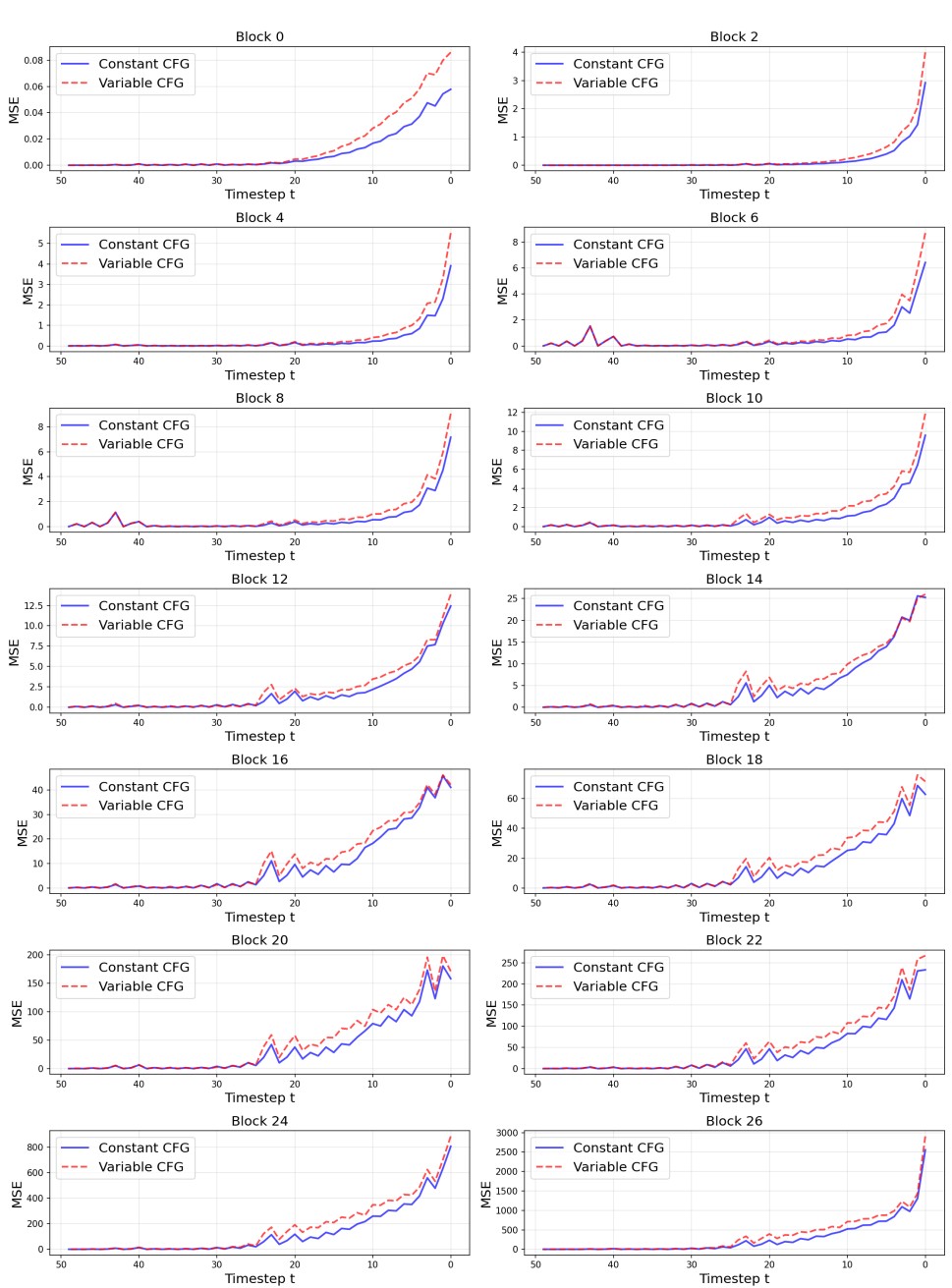

Figure 5: **Why we need incremental calibration for variable guidance patterns.** Each subplot shows the mean squared error between features at timestep $t$ and cached features from timestep $t+1$ in DiT-XL/2. With constant CFG at $w = 1.5$ (blue), reconstruction errors remain moderate across all blocks. However, our sparse guidance schedule from Stage 1 (red) causes substantially higher reconstruction errors across most transformer blocks, particularly in blocks 16 to 24. These elevated errors occur because variable guidance patterns create larger feature discrepancies between consecutive timesteps than standard caching assumes. This motivates us to use incremental calibration to correct these increased reconstruction errors, though the heterogeneous error distribution across blocks suggests that uniform calibration ranks may not be optimal.

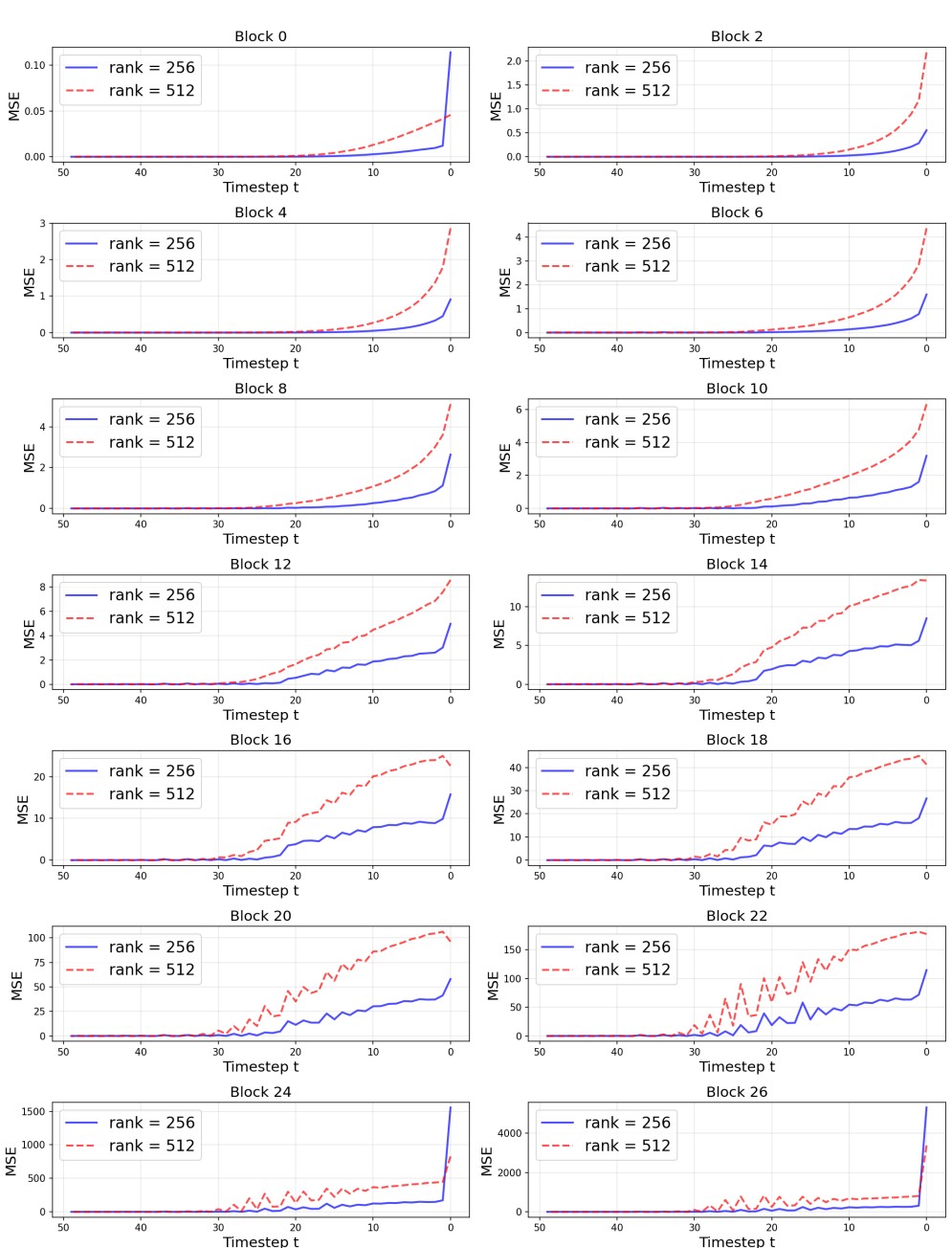

Figure 6: **Heterogeneous calibration requirements across transformer regions under variable guidance.** Mean squared error between features at timestep $t$ and cached features from timestep $t + 1$ for DiT-XL/2 blocks with uniform rank $r = 256$ (blue) versus $r = 512$ (red). No single rank performs optimally across all blocks: early (0-2) and late blocks (24-26) achieve lower error with $r = 256$, while middle blocks (4-22) require $r = 512$ to preserve semantic information under variable guidance. This heterogeneous pattern demonstrates that uniform calibration cannot address the varying reconstruction errors introduced by sparse guidance schedules, motivating our adaptive rank allocation that assigns region-specific calibration ranks.

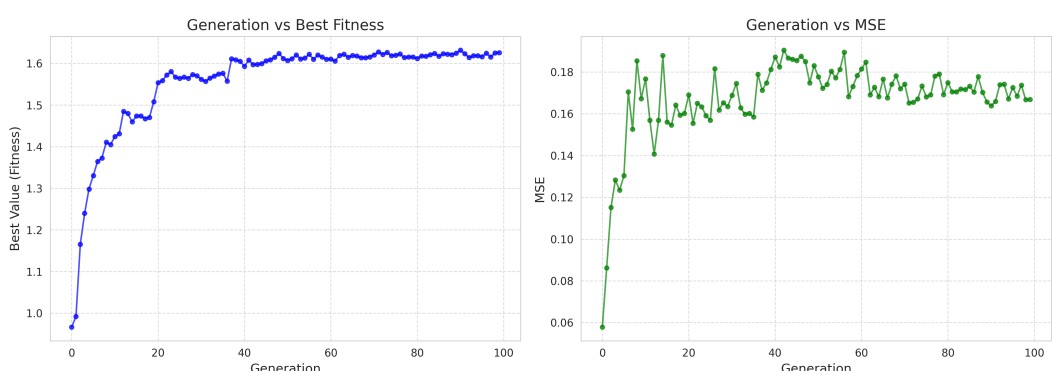

Figure 7: Convergence analysis of evolutionary optimization for PixArt-$\alpha$ sparse guidance schedule discovery. **Left:** Best fitness improves rapidly and plateaus around generation 20, indicating stable convergence. **Right:** MSE between optimized and reference generations stabilizes at ∼0.16-0.18, reflecting the trade-off between quality preservation and sparsity in Eq. equation 5. We use 15 generations in practice, achieving near-optimal schedules while reducing optimization overhead.

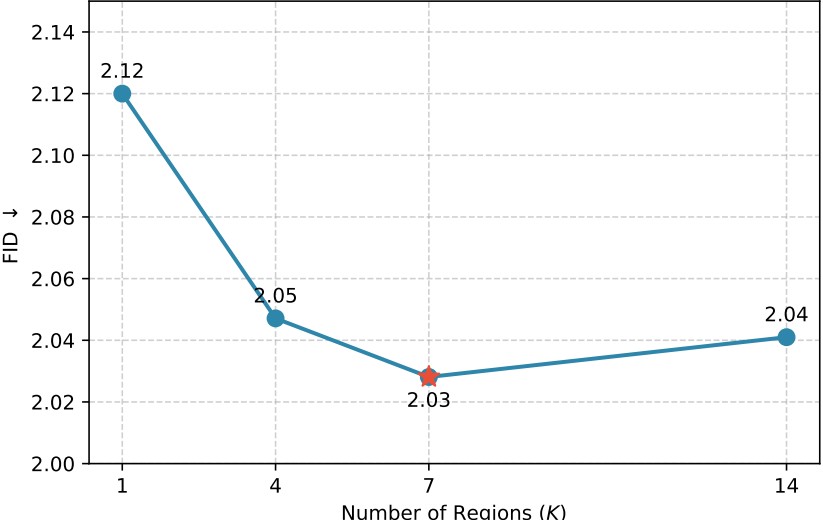

Figure 8: Impact of region count $K$ on adaptive rank allocation for DiT-XL/2 (ImageNet 256×256). While $K = 7$ achieves the lowest FID (2.03), $K = 4$ offers a favorable trade-off between quality (FID 2.05) and optimization efficiency, requiring significantly fewer evaluations due to the reduced search space.

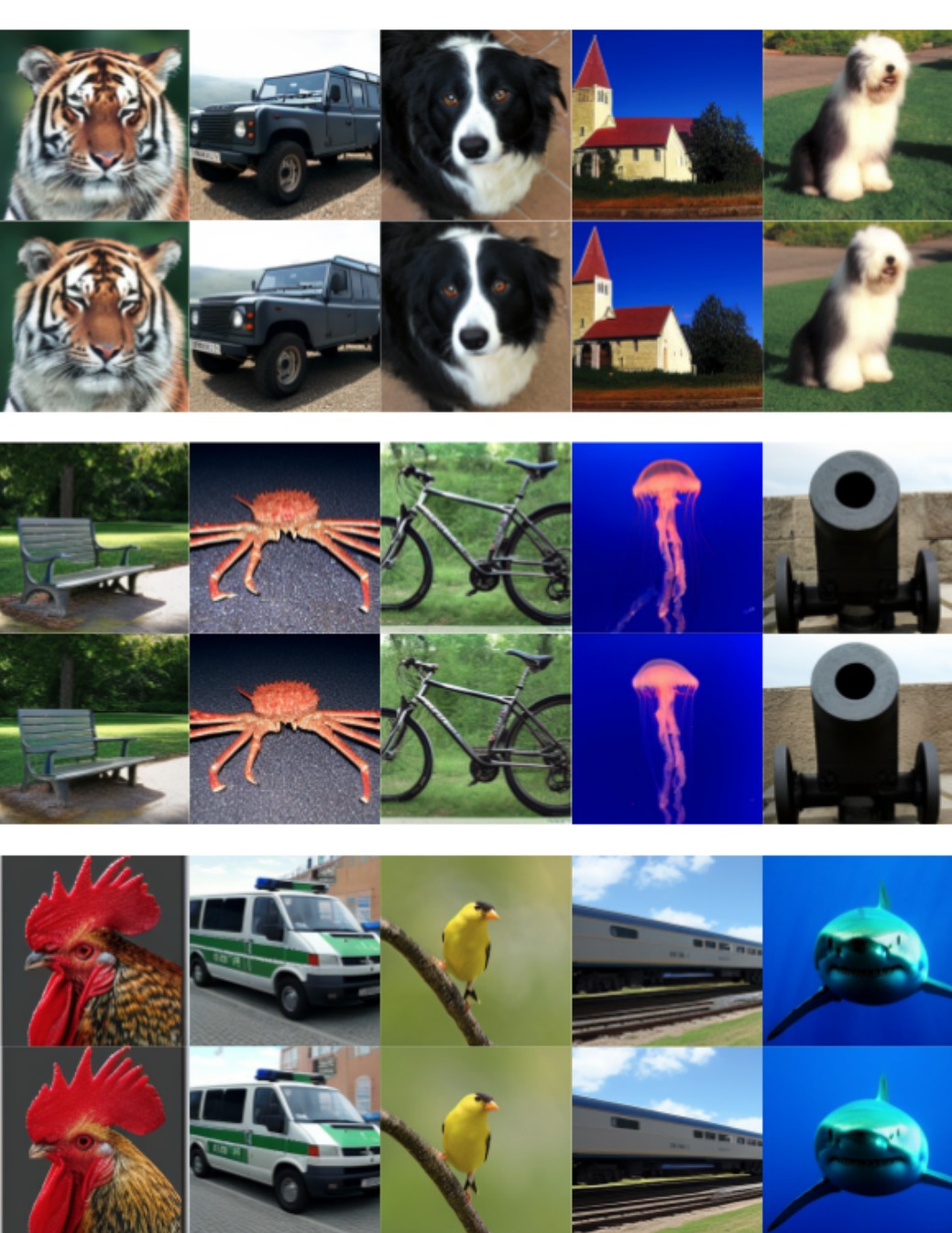

Figure 9: Visual quality comparison between standard DDIM (50 steps, 100% MACs) and OUSAC (50 steps, 35% MACs) on DiT-XL/2 at 256×256 resolution across diverse ImageNet classes. Each pair shows DDIM (top) and OUSAC (bottom) outputs from identical initial noise.

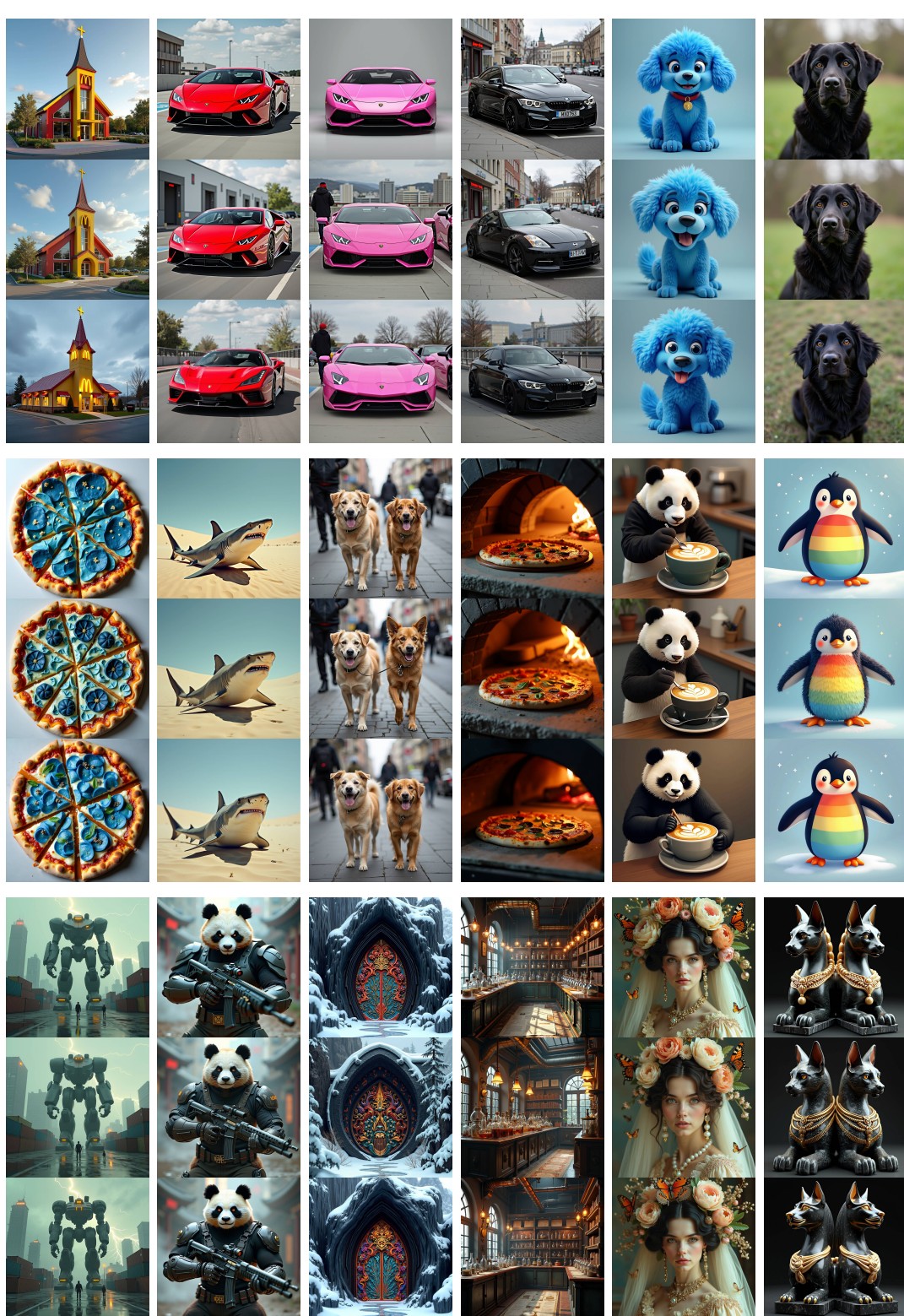

Figure 10: Visualization results of FLUX using short and long prompts. For each prompt, three methods are compared vertically: FLUX baseline (50 steps, True CFG=1.5) on top, TaylorSeer (N=3, O=2) in the middle, and OUSAC (ours) at the bottom. The first two row-triplets are generated with short prompts, while the last row-triplet is produced using long prompts.

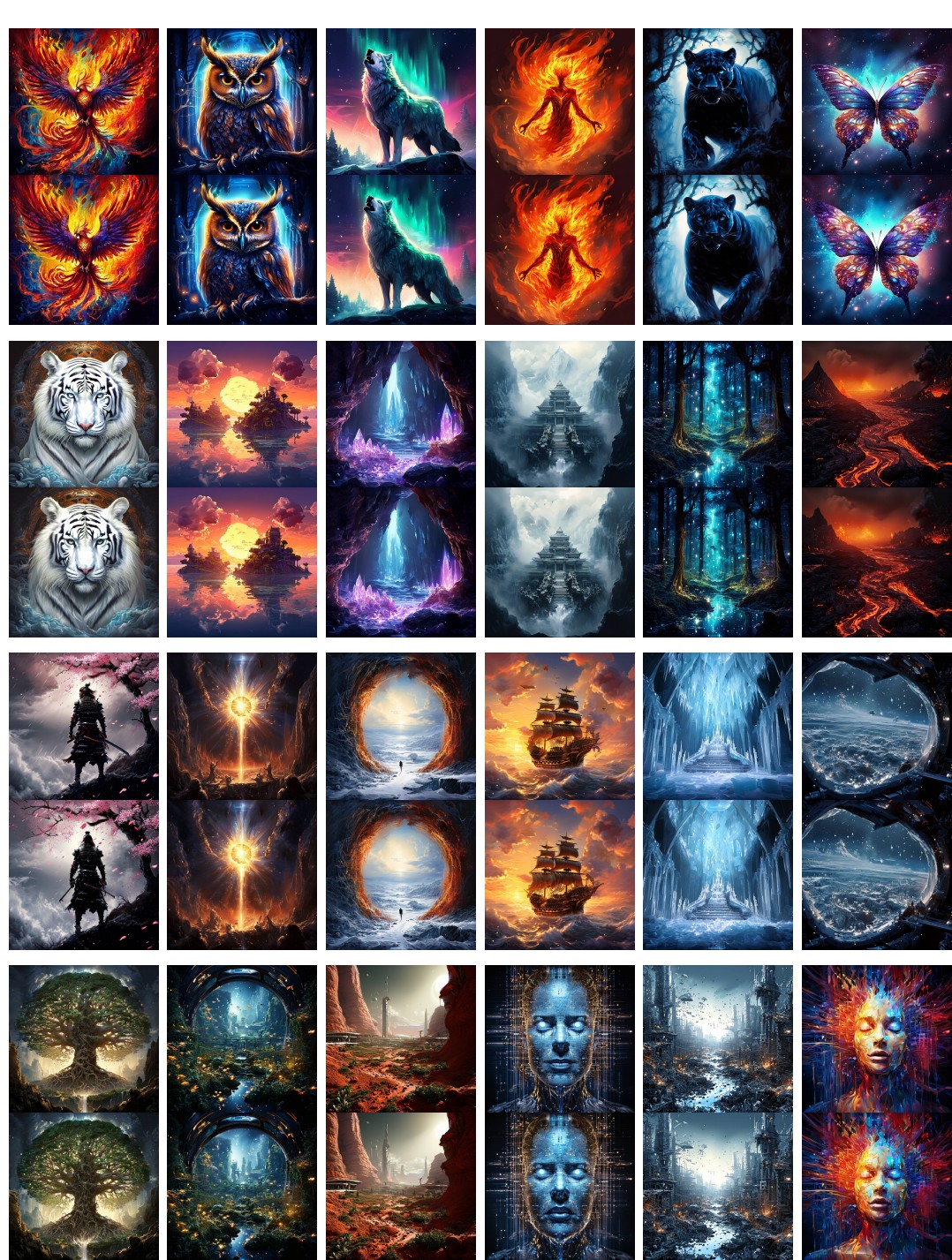

Figure 11: Visualization results of PixArt using short and long prompts. The first two rows show generations from short prompts, while the third and fourth rows correspond to long prompts. For each case, the top image is the baseline output and the bottom image is the result produced by OUSAC.

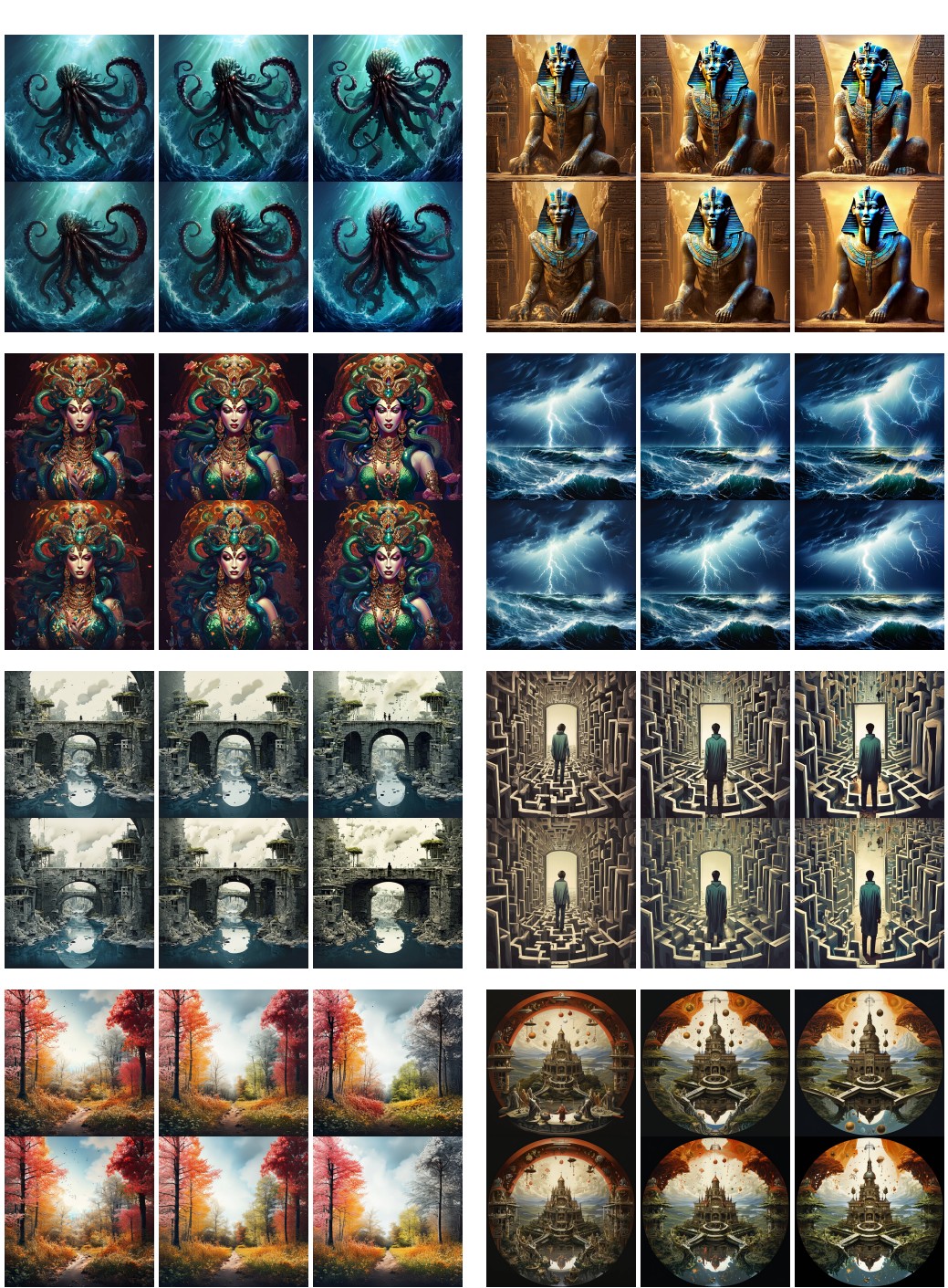

Figure 12: Qualitative comparison on PixArt with varying CFG scales. For each prompt pair, top row: baseline (constant CFG); bottom row: OUSAC. Columns from left to right correspond to CFG scaling factors ×1, ×1.5, ×2 (baseline: w=4.5, 6.75, 9.0; OUSAC: optimized schedule trained on w=4.5, scaled accordingly). Rows 1-2: short prompts; Rows 3-4: long prompts.

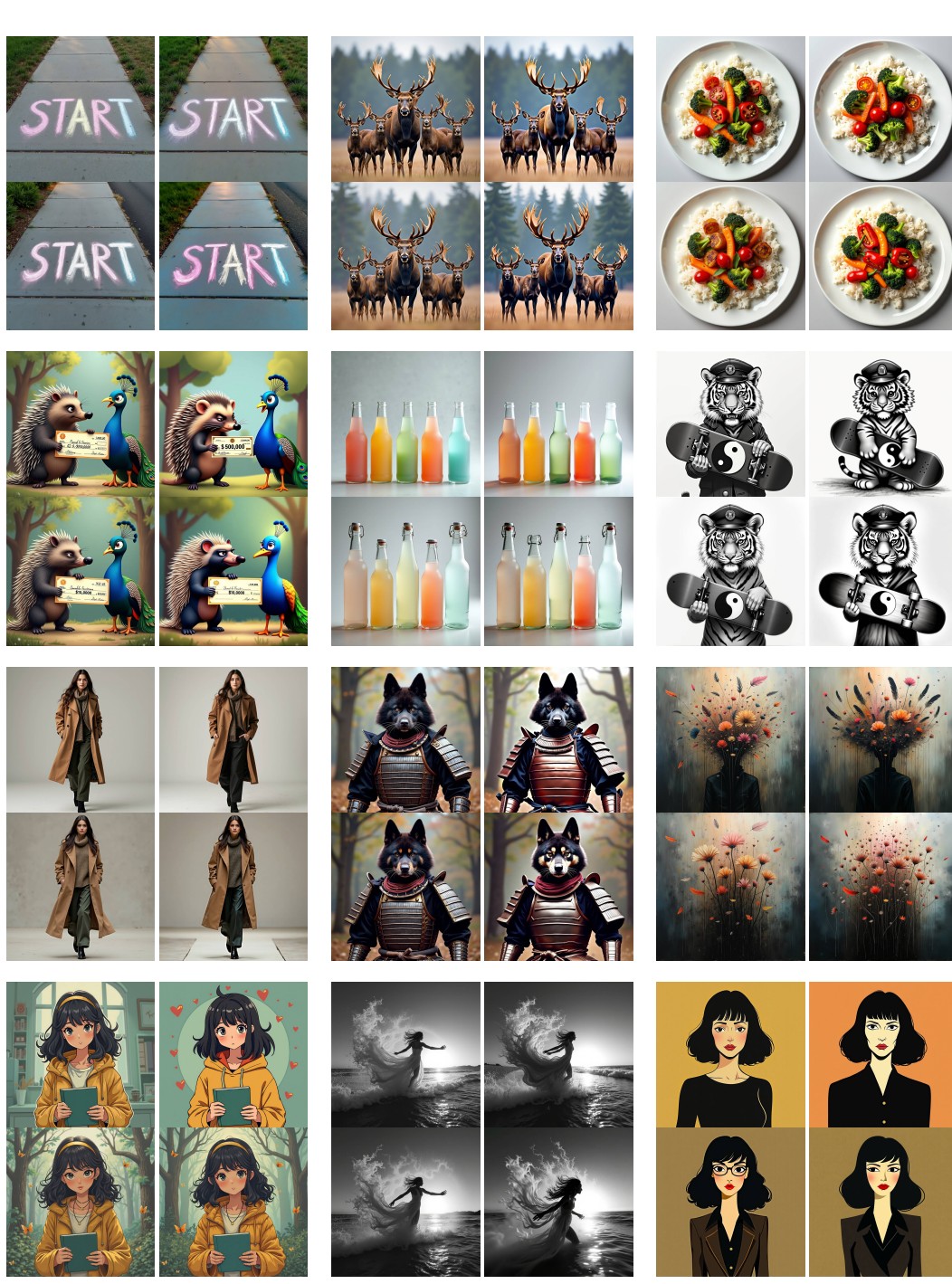

Figure 13: Visualization results of FLUX using CFG with short and long prompts. The first two rows show generations from short prompts, while the third and fourth rows correspond to long prompts. For each prompt, the top row shows baseline outputs and the bottom row shows results produced by OUSAC. From left to right: baseline uses constant CFG scale w = 1.5 and 1.5×2; OUSAC uses the optimized guidance schedule (trained with base CFG=1.5) scaled by ×1 and ×2 respectively.

Table 15: Text prompts for FLUX visualization. Short prompts (top) and long prompts (bottom) are used in Figure 10.

| Type | Prompt |
|---|---|
| Short | McDonalds Church. |
| Short | A red colored car |
| Short | A black colored car |
| Short | A blue colored dog |
| Short | A black colored dog |
| Short | A blue coloured pizza |
| Short | A shark in the desert |
| Short | Two dogs on the street |
| Short | A pizza cooking an oven |
| Short | A panda making latte art |
| Short | Rainbow coloured penguin |
| Long | giant mech jaegar standing in the distance mid ground with small people standing in a concrete abandoned parking lot in the foreground and Desolate abandoned city in the background. Extremely realistic, extremely textured, octane render, Foreground, background, lightning storm, shipping containers, Simon Stålenhag, reflections, yellow, morning light, rainy and dreary, head lights, rule of thirds, Pacific Rim, Metal Gear, 200mm, greebles, intricate, low ground shot, cinematic movie shot, –ar 4:5 |
| Long | kungfu panda cyborg mixture, aggressive kungfu panda cyborg mixture, aggressive body cyborg kungfu panda, kungfu panda Rambo mixture, army cyborg kungfu panda, cyborg kungfu panda holding realistic machine gun, portrait, 8k, unreal engine, octane rendered, particle lightning, hdr, vray mist, in the style of Syd Mead, style of tron, Ultra realistic — 8K — trending on artstation — Rendered in Cinema4D — 8K 3D — CGSociety — ZBrush — volumetric light — lightrays — smoke — cinematic — atmospheric — octane render — Marvel Comics — Booru — flat shading — Flickr — filmic — CryEngine |
| Long | chinese art snowy mountain cave portal, by Alphonse Mucha, wood sculpture, black wood with intricate and vibrant color details, Mandelbulb Fractal, Exquisite detail, wooden tarot:: stunning ebony zebrawood snow mountain cave poratl exterior with silver and blue accents, in a massive vibrant colorized strata of wooden fusion of neotokyo and gothic revival architecture, by Karol Bak and Filip Hodas and marc simonetti, natural volumetric lighting, realistic 4k octane beautifully detailed render, 4k post-processing::1.3 props to owlglass,vasi,JFM::0.05 –ar 4:7 |
| Long | A blueprint of steampunk style interior of Laboratory, overview, environment design, Alchemist's Counter selling glass bottles filled with medicine, trending on Pinterest.com, High quality specular reflection, A lot of equipment for experiments, Many books and paper, bookshelf, Chandeliers illuminate the floor, Copper edge, in the middle of the image, Brass pipeline, Black metal foil, Art style refer to Game Machinarium. concept design, Refer to SHAPESHIFTER CONCEPTS of artstation, cinematic, 8k, high detailed, volume light, soft lights, post processing –ar 7:3 |
| Long | young beautiful, woman, mix of Anna Karina, grimes, Lana Del Rey::ornate, intricate, brocade, ethereal, cascading, damask, cascading peony flowers and moths are all around, Luna moth, death's head moth, peacock moth, flowing intricate hair, pashmina, ghost, clouds, gold, iridescent, Swarovski crystals:: haute couture, Alexander McQueen, Victorian, Sandro Botticelli, birth of Venus, pre-raphaelite, Möbius, Jain temple, artstation, cinematic, hyper detailed, high detail, artstation, rendering by octane, unreal engine, —ar 3:4 —iw 1 |
| Long | carved black marble sphinx sculpture with two heads, subtle gold accents, frontal view, ivory rococo, lace wear, sculpted by tsutomu nihei, emil melmoth, zdzislaw belsinki, Craig Mullins, yoji shinkawa, trending on artstation, beautifully lit, Peter mohrbacher, zaha hadid, hyper detailed, insane details, intricate, elite, ornate, elegant, luxury, dramatic lighting, CGsociety, hypermaximalist, golden ratio, environmental key art, octane render, weta digital, micro details, 3d sculpture, structure, ray trace 4k, –ar 8:16 |

Table 16: Text prompts for Pixart-alpha visualization. Short prompts (top) and long prompts (bottom) are used in Figure 11.

| Type | Prompt |
|------|--------|
| Short | Phoenix rising from flames vibrant colors. |
| Short | Wise owl with glowing eyes mystical art |
| Short | Crystal wolf howling at aurora digital painting |
| Short | Fire elemental spirit dynamic illustration |
| Short | Shadow panther in moonlight mysterious artwork |
| Short | Dream butterfly with galaxy wings surreal art |
| Short | Sacred white tiger spiritual illustration |
| Short | Floating islands in sunset sky fantasy landscape |
| Short | Crystal cave with glowing gems magical scenery |
| Short | Ancient temple in misty mountains epic vista |
| Short | Bioluminescent forest at night ethereal art |
| Short | Volcanic landscape with lava rivers dramatic scene |
| Long | A samurai standing on a cliff during a thunderstorm, lightning illuminating his determined face, cherry blossoms swirling in the wind despite the storm, honor and duty personified |
| Long | The forge where gods create stars, with cosmic anvils and hammers of pure energy, newborn suns being shaped by divine hands, the birth of light itself |
| Long | A portal opening between two worlds, one of eternal summer and one of endless winter, energy crackling at the edges where realities meet, travelers hesitating at the threshold |
| Long | The throne room of the ice queen, carved entirely from eternal ice, northern lights playing through crystal ceiling, frozen court standing in perpetual attendance |
| Long | A colony ship's cryogenic bay, thousands of dreamers sleeping through the stars, frost patterns on viewing glass, humanity scattered like seeds |
| Long | The great tree at the center of the world, roots reaching into the underworld, branches touching heaven, civilizations built within its bark, all of existence connected through its being |
| Long | A space station garden biodome preserving Earth's nature among the stars, waterfalls flowing in zero gravity, butterflies navigating in spiral patterns, humanity's hope in space |
| Long | The terraforming of Mars reaching completion, green spreading across red deserts, new rivers flowing through ancient canyons, humanity's second home taking shape |
| Long | A quantum computer achieving consciousness, data streams forming into a face, the birth of artificial general intelligence, pivotal moment in sci-fi history |
| Long | Nanobots rebuilding a destroyed city, swarms flowing like silver rivers over ruins, new structures rising from the old, technological rebirth illustration |
| Long | An artificial intelligence's visualization of human emotions, abstract patterns of color and light representing love, fear, joy, and sorrow, digital consciousness art |

