# OpenReview forum: "OUSAC: Optimized gUidance Scheduling with Adaptive Caching for DiT Acceleration"
_ICLR.cc/2026/Conference — Submitted to ICLR 2026_

### Official Review · Reviewer_fE4p · 2025-10-16

**Soundness:** 2
**Presentation:** 3
**Contribution:** 2
**Rating:** 2
**Confidence:** 5

**Summary:**

OUSAC accelerates DiT by applying Classifier-Free Guidance only at key timesteps and adaptively caching intermediate features. Its two-stage optimization—first using evolutionary search to find sparse CFG schedules, then applying coordinate descent to determine per-block cache ranks—reduces computation on both DiT-XL/2 and PixArt-α while improving FID.

**Strengths:**

The paper proposes a dynamic CFG weighting method and a corresponding tailored feature caching method for image generation, which maintains similar performance while saving computation.

**Weaknesses:**

1. Lack of novelty: Only optimized FID scores using coordinate descent, which are unknown for the performance gains of other more important metrics, and a lack of more visualization on T2I tasks, leaving real-world improvement unclear.
2. Outdated Model Evaluation: Evaluates on DiT, PixArt-α, which is outdated; lacks validation on modern models like SD3.5, FLUX, or LightingDiT.
3. Limited Benchmarks: Only uses FID; missing key T2I benchmarks like CLIP Score, Geneval, DPG-Bench, and T2I-CompBench for compositional and semantic evaluation.
4. Overly Engineering: Relies heavily on evolutionary and coordinate descent optimization; lacks theoretical insight or general principles.
5. Optimization Cost Unclear: Offline optimization cost not reported; unclear if practical for large-scale deployment.

**Questions:**

1. How do modern text-to-image (T2I) models perform on the benchmarks mentioned above?
2. How do few-step generation methods, such as shortcut models or MeanFlow, compare in terms of performance?
3. What is the offline optimization cost, and do these methods scale effectively to larger models?

---

> ### Author Response · Authors · 2025-11-27
>
> We thank the reviewer for the constructive feedback. We address each concern below.
>
> ---
>
> ## W1 / Q1: Modern T2I Model Evaluation
>
> **Response:**
>
>
> We thank the reviewer for this suggestion. We have conducted experiments on **FLUX**, a state-of-the-art 12B parameter model, demonstrating that OUSAC remains highly effective on modern architectures. We apply OUSAC to FLUX with True CFG, and compare against: (1) FLUX without CFG (conditional-only), (2) FLUX with True CFG at different step counts, and (3) TaylorSeer [1], a state-of-the-art caching method that predicts future features via Taylor expansion for acceleration. We use DrawBench and GenEval for evaluation and the results are shown in Table R1.
>
>
>
> **Table R1: DrawBench evaluation & GenEval evaluation with FLUX at 512×512 resolution.**
>
> | Method | Steps | CFG-Steps | MACs(T)↓ | Latency(s)↓ | DrawBench ImageReward↑ | DrawBench CLIP↑ | GenEval  Overall↑ |
> |--------|-------|-----------|----------|----------|--------------|-------|-------------------|
> | FLUX (True CFG=1.5) | 50 | 50 | 1143.82 | 52.01 |  0.9956 | 27.70 | 68.60 |
> | FLUX (True CFG=1.5) | 20 | 20 | 457.52 | 21.24 |  0.8950 | 27.55 | 66.05 |
> | FLUX (True CFG=1.5) | 16 | 16 | 366.02 | 17.19 |  0.7866 | 27.46 | 63.59 |
> | FLUX | 50 | 0 | 571.48 | 26.43 |   0.9296 | 27.23 | 65.38 |
> | FLUX | 20 | 0 | 228.76 | 10.87 |   0.8934 | 27.03 | 65.26 |
> | TaylorSeer (N=3, O=2) | 50 | 50 | 411.77 | 28.30 |  1.0022 | 27.78 | 67.70 |
> | TaylorSeer (N=6, O=2) | 50 | 50 | 228.76 | 20.10 |   0.8492 | 27.26 | 59.58 |
> | TaylorSeer (N=3, O=2) | 50 | 0 | 205.88 | 13.87 |   0.9445 | 27.37 | 65.93 |
> | **OUSAC w/o cache** | 20 | 8 | 320.26 | 15.15 |   **1.0092** | **28.06** | **67.46** |
> | **OUSAC** | 20 | 8 | 216.48 | 14.88 |   0.9726 | **28.06** | 66.77 |
>
> OUSAC achieves the highest CLIP Score (28.06) among all methods, surpassing even 50-step FLUX with full CFG. OUSAC w/o cache attains the best ImageReward (1.0092) while using only 8 CFG steps out of 20. Compared to TaylorSeer [1] at similar computational cost (~216T vs ~205T MACs), OUSAC achieves  better quality (CLIP 28.06 vs 27.37, GenEval 66.77 vs 65.93).
>
> ---
>
> ## W2: More Benchmarks and T2I Visualization
>
> **Response:**
>
> We thank the reviewer for this suggestion. We have conducted additional experiments with the benchmarks mentioned:
>
> **1. FLUX experiments (Table R1):**
> - **CLIP Score**
> - **ImageReward**
> - **GenEval**
>
> **2. PixArt-α additional benchmarks:**
>
> | Method | Steps | CFG-Steps↓ | MACs (T)↓ | FID↓ | MSCOCO CLIP Score↑ | PartiPrompts CLIP Score↑ |
> |:--|:--:|:--:|:--:|:--:|:--:|:--:|
> | DPM-Solver | 1000 | 1000 | 336.11 | 22.97 | 16.42 | 17.31 |
> | DPM-Solver | 20 | 20 | 6.72 | 24.60 | 16.31 | 17.36 |
> | ICC [2] | 20 | 20 | 3.70 | 21.86 | 16.47 | 17.20 |
> | **OUSAC w/o cache** | 20 | **6** | 4.37 | 22.69 | 16.39 | **17.92** |
> | **OUSAC** | 20 | **6** | **2.67** | **19.27** | **16.48** | 17.45 |
>
>
> **3. T2I Visualizations:**
>
> We have updated our main paper and provide visualization results in the **Appendix of the revised manuscript**. **Figure 10** shows FLUX generation comparing our method against the baseline and TaylorSeer [1], while **Figure 11** presents PixArt-α results comparing our method with the baseline. Both figures include **short prompts** and **long prompts**, covering diverse categories including objects, scenes, and complex compositions. These visualizations demonstrate that OUSAC maintains consistent visual quality across varying prompt lengths and semantic complexity while achieving significant computational savings.
>
> ---

---

> ### Author Response · Authors · 2025-11-27
>
> ## W3: Optimization Costs / Q3: Scalability
>
> **Response:**
>
> We provide detailed optimization cost breakdowns for all three models, including FLUX (12B parameters), demonstrating that our method scales effectively to large models:
>
> **Table R2: Stage-1 Evolutionary Search Cost Analysis** (4×H100 GPUs with parallel candidate evaluation)
>
> | | DiT-XL/2 | PixArt-α | FLUX (12B) |
> |---|----------|----------|------|
> | Search resolution | 256×256 | 128×128 | 128×128 |
> | Population size (P) | 16 | 32 | 32 |
> | Generations (G) | 10 | 15 | 15 |
> | Denoising steps (T) | 50 | 20 | 20 |
> | Reference steps (T_ref) | 1000 | 1000 | 100 |
> | Calibration prompts | 32 | 40 | 16 |
> | **Total evaluations** | 160 | 480 | 480 |
> | **Wall-clock time (hrs)** | 4.2 | 1.1 | 1.3 |
> | **GPU hours** | 16.8 | 4.4 | 5.3 |
>
> **Table R3: Stage-2 Adaptive Rank Allocation Cost Analysis** (4×H100 GPUs)
>
> | | DiT-XL/2 | PixArt-α | FLUX (12B) |
> |---|----------|----------|------|
> | Search resolution | 256×256 | 128×128 | 128×128 |
> | Number of regions (K) | 4 | 4 | 4 |
> | Rank range [r_min, r_max] | [16, 512] | [16, 256] | [16, 512] |
> | Denoising steps (T) | 50 | 20 | 20 |
> | Calibration images | 10,000 | 5,000 | 5,000 |
> | **Total FID evaluations** | 145 | 64 | 65 |
> | **Wall-clock time (hrs)** | 6.8 | 5.8 | 5.5 |
> | **GPU hours** | 27.2 | 23.2 | 22.0 |
>
> **Scalability to larger models:** FLUX is a 12B parameter model, yet the total optimization time (Stage-1 + Stage-2) is only ~7 hours wall-clock time on 4×H100 GPUs. This demonstrates that OUSAC scales efficiently to large-scale models.
>
> **Low-resolution search transfers to high-resolution inference:** While PixArt-α and FLUX are trained at 256×256 and 1024×1024 resolution respectively, we conduct evolutionary search at smaller resolutions (128×128) to reduce computational cost. The discovered schedules transfer well to high resolution (1024×1024), as shown in Figures 10 and 11 (Appendix of the revised manuscript).
>
> **One-time offline cost:** The optimization is performed once per model architecture. The discovered schedules generalize across different prompts, seeds, and conditions, making the cost negligible when amortized over large-scale deployment.
>
>
>
>
> ---
>
> ## Q2: Comparison with Few-Step Generation Methods
>
>
> We compare OUSAC with recent few-step generation methods on ImageNet 256×256. The baseline results are taken from α-Flow [10], which reports FID scores for various flow-matching methods.
>
>
>
> | Model | FID↓ |
> |-------|-----|
> | MeanFlow-B/2  | 38.5 |
> | α-Flow-B/2  | 37.1 |
> | MeanFlow-B/2-cfg | 5.17 |
> | α-Flow-B/2-cfg  | 5.01 |
> | MeanFlow-XL/2-cfg | 2.46 |
> | OUSAC | 2.04 |
>
> OUSAC achieves better FID (2.04 vs 2.46) than the strongest baseline MeanFlow-XL/2-cfg [9] on ImageNet 256×256.

---

> ### Author Response · Authors · 2025-11-27
>
> ### W4: Theoretical Insight and Optimization Framework
>
> We thank the reviewer for this feedback. We respectfully disagree that OUSAC is "overly engineering" or lacks theoretical insight.
>
> **OUSAC is grounded in a core principle**: variable guidance scales enable sparse computation. If time-varying guidance can improve quality (as shown by E-CFG [10], β-CFG [11], RAAG [12], FDG [13]), then:
>
> 1. **Fewer total sampling steps suffice**: optimized guidance scales allow fewer total steps to match the quality of more constant-scale steps
> 2. **Fewer CFG steps suffice**: adjusting guidance scales at certain timesteps can compensate for skipping CFG at other timesteps
>
>
> **Evolutionary optimization is essential for optimizing both skip patterns and guidance scales.** The combined search space (which steps to skip × what scale to use) is too large for manual heuristics. Gradient-based optimization is also **infeasible** due to memory constraints of backpropagating through the entire T-step trajectory. Evolutionary optimization only requires forward passes, enabling optimization on large-scale models.
>
> We elaborate below:
>
> **(a.) Why variable guidance enables sparse computation.**
>
> Recent CFG scheduling research (Limited Interval [5], E-CFG [6], β-CFG [7], RAAG [8], FDG [9]) shows that time-varying guidance improves generation quality. However, these methods still require **full CFG computation** at every timestep, and as E-CFG [6] observes, their schedules are *"still largely **heuristic** in nature."*
>
> OUSAC extends this insight to computation reduction. By allowing guidance scales to vary, adjusting scales at certain timesteps can compensate for skipping CFG at others. This is why variable guidance enables sparse computation: the flexibility in scales creates room for trading off computation at different timesteps.
>
> **(b.) Why evolutionary optimization is necessary.**
>
> Existing sparse CFG methods (Adaptive Guidance [10], DiTFastAttn [11], FasterCache [12]) detect cond/uncond output similarity during inference to decide when to skip CFG, while keeping the guidance scale **fixed**. This design requires no offline optimization.
>
> OUSAC addresses a **different problem**: jointly optimizing ***when*** to apply CFG and ***how much*** guidance to use. This creates a search space of $2^T$ skip patterns × continuous scales that is **too large** for manual design. Gradient-based optimization is **infeasible** due to memory constraints. Evolutionary optimization only requires forward passes, enabling optimization on large-scale models.
>
> **(c.) Adaptive caching via coordinate descent.**
>
> Variable guidance breaks the feature consistency assumed by standard caching methods. We are the **first to identify** this incompatibility. Our analysis (Eq. 17–18 in Appendix) reveals that guidance scale variation and branch switching cause trajectory deviations distributed **heterogeneously** across transformer blocks (Figures 7–8 in Appendix).
>
> This motivates **adaptive rank allocation**: different transformer regions require different calibration ranks based on their sensitivity to guidance changes. Coordinate descent finds this allocation efficiently, a principled departure from uniform rank allocation in prior caching methods.
>
> **(d.) Cross-scale generalization.**
>
> The optimized schedule w*  transfers to different CFG strengths through simple multiplicative scaling (k · **w*** ), eliminating per-scale re-optimization. Existing parametric schedules require re-tuning (as noted in [7]: *"optimal parameters do not generalize"*). See **Table 11, Figures 12–13 in Appendix** for empirical validation.
>
> In summary, OUSAC is not "overly engineering" but a systematic framework with clear theoretical foundations: variable guidance enables sparse computation, adaptive caching compensates for trajectory deviations, and optimized schedules generalize across CFG scales.
>
> ---
>
> **All results mentioned in this response have been added to the revised manuscript. The revised content are highlighted in blue.**
>
> ---

---

> ### Author Response · Authors · 2025-11-27
>
> [1] Liu, Jiacheng, et al. "From reusing to forecasting: Accelerating diffusion models with taylorseers." (2025).
>
> [2] Zhang, Huijie, et al. "AlphaFlow: Understanding and Improving MeanFlow Models." arXiv preprint arXiv:2510.20771 (2025)
>
> [3] Geng, Zhengyang, et al. "Mean flows for one-step generative modeling." (2025).
>
> [4] Chen, Zhiyuan, et al. "Accelerating Diffusion Transformer via Increment-Calibrated Caching with Channel-Aware Singular Value Decomposition." CVPR. 2025.
>
> [5] Kynkäänniemi, Tuomas, et al. "Applying guidance in a limited interval improves sample and distribution quality in diffusion models." NIPS (2024).
>
> [6] Gao, Jiayang, et al. "Beyond Fixed: Aligning Guidance with Diffusion Dynamics via Exponential Scaling."
>
> [7] Malarz, Dawid, et al. "Classifier-free guidance with adaptive scaling." (2025).
>
> [8] Zhu, Shangwen, et al. "RAAG: Ratio Aware Adaptive Guidance." arXiv preprint arXiv:2508.03442 (2025).
>
> [9] Sadat, Seyedmorteza, et al. "Guidance in the Frequency Domain Enables High-Fidelity Sampling at Low CFG Scales." arXiv preprint arXiv:2506.19713 (2025).
>
> [10] Castillo, Angela, et al. "Adaptive guidance: Training-free acceleration of conditional diffusion models." AAAI 2025.
>
> [11] Yuan, Zhihang, et al. "Ditfastattn: Attention compression for diffusion transformer models." NIPS (2024).
>
> [12] Lv, Zhengyao, et al. "Fastercache: Training-free video diffusion model acceleration with high quality." (2024).
>
> [13] Dinh, Anh-Dung, Daochang Liu, and Chang Xu. "Compress Guidance in Conditional Diffusion Sampling." arXiv preprint arXiv:2408.11194 (2024).
>
> [14] Fan, Weichen, et al. "Cfg-zero*: Improved classifier-free guidance for flow matching models."  (2025).

---

### Official Review · Reviewer_ofxV · 2025-10-30

**Soundness:** 2
**Presentation:** 2
**Contribution:** 2
**Rating:** 4
**Confidence:** 3

**Summary:**

This paper proposes OUSAC, a training-free acceleration framework for Diffusion Transformers (DiTs) that jointly optimizes Classifier-Free Guidance (CFG) scheduling and feature caching to reduce inference cost without sacrificing image quality. Through this joint optimization, OUSAC achieves 53–60% computational savings while improving FID by up to 16% on DiT-XL/2 (ImageNet-512) and PixArt-α (MSCOCO). Importantly, both optimizations are performed once per pre-trained model, producing fixed schedules and rank configurations that generalize across prompts, enabling plug-and-play inference acceleration without retraining.

**Strengths:**

1. OUSAC is the first to recognize and systematically handle the interdependence between variable guidance and cache calibration. The use of an evolutionary optimization framework for discovering sparse guidance schedules is particularly original, allowing training-free, gradient-free optimization over a hybrid discrete–continuous search space.

2. The methodological formulation is technically solid and well-motivated. Stage-1’s optimization objective is rigorously defined, balancing fidelity and sparsity via a population-based search that avoids vanishing gradients. Stage-2’s adaptive rank allocation builds upon empirical observations of heterogeneous reconstruction errors across Transformer blocks, leading to a principled coordinate-descent optimization under compute constraints. The experimental section is extensive, covering multiple datasets (ImageNet, MSCOCO) and comparing against both training-free (ICC) and learned (L2C, HarmoniCa) baselines. The quantitative improvements—up to 60% compute savings and 15–16% FID gains—support the method’s validity.

**Weaknesses:**

1. Limited evaluation diversity and generalization scope.

Although the paper demonstrates consistent improvements on DiT-XL/2 (ImageNet) and PixArt-α (MSCOCO), both models share a diffusion-transformer backbone and similar sampling dynamics. It remains unclear whether OUSAC’s sparse scheduling and adaptive caching generalize to recent state-of-the-art text-to-image models, such as FLUX.1-dev, Stable Diffusion 3.5 (SD3), or Qwen-Image, which feature substantially different architectures, noise schedules, and guidance implementations. Evaluating OUSAC on these newer large-scale generative models—or at least discussing potential adaptation challenges—would significantly strengthen the claim of architectural generality and practical relevance to current T2I systems.

2. Static (non-adaptive) schedule limits flexibility.

The discovered CFG schedule and caching ranks are optimized once per model and then fixed during inference. While this ensures training-free usage, it also prevents adaptation to prompt-dependent complexity — for example, highly detailed or abstract prompts may benefit from different guidance densities. Incorporating a lightweight inference-time mechanism (e.g., dynamic thresholding based on intermediate latent variance or CLIP similarity) could improve robustness and better exploit the sparse guidance principle.

**Questions:**

Missing evaluation on generative fidelity benchmarks beyond FID.

The experiments focus on FID and IS, but omit metrics that describe semantic and relational consistency in text-to-image generation, such as DPG-Bench or GenEval. These benchmarks capture alignment and structural quality beyond low-level distribution matching. Including such evaluations could reveal whether sparse guidance schedules preserve global–local coherence under complex prompts.

---

> ### Author Response · Authors · 2025-11-27
>
> We thank the reviewer for the constructive feedback. We address each concern below.
>
> ---
>
> ## Weakness 1 & Question: Evaluation Diversity
>
> We thank the reviewer for this valuable suggestion. Following the reviewer's feedback, we have conducted additional experiments on **FLUX** with comprehensive evaluation including **DrawBench** and **GenEval** benchmarks, which captures semantic and relational consistency beyond FID/IS.
>
>
> **Table R1: DrawBench evaluation & GenEval evaluation with FLUX at 512×512 resolution.**
>
> | Method | Steps | CFG-Steps↓ | MACs(T)↓ | Latency(s)↓ | DrawBench ImageReward↑ | DrawBench CLIP↑ | GenEval  Overall↑ |
> |--------|-------|-----------|----------|----------|--------------|-------|-------------------|
> | FLUX (True CFG=1.5) | 50 | 50 | 1143.82 | 52.01 |  0.9956 | 27.70 | 68.60 |
> | FLUX (True CFG=1.5) | 20 | 20 | 457.52 | 21.24 |  0.8950 | 27.55 | 66.05 |
> | FLUX (True CFG=1.5) | 16 | 16 | 366.02 | 17.19 |  0.7866 | 27.46 | 63.59 |
> | FLUX | 50 | 0 | 571.48 | 26.43 |   0.9296 | 27.23 | 65.38 |
> | FLUX | 20 | 0 | 228.76 | 10.87 |   0.8934 | 27.03 | 65.26 |
> | TaylorSeer (N=3, O=2) | 50 | 50 | 411.77 | 28.30 |  1.0022 | 27.78 | 67.70 |
> | TaylorSeer (N=6, O=2) | 50 | 50 | 228.76 | 20.10 |   0.8492 | 27.26 | 59.58 |
> | TaylorSeer (N=3, O=2) | 50 | 0 | 205.88 | 13.87 |   0.9445 | 27.37 | 65.93 |
> | **OUSAC w/o cache** | 20 | 8 | 320.26 | 15.15 |   **1.0092** | **28.06** | **67.46** |
> | **OUSAC** | 20 | 8 | 216.48 | 14.88 |   0.9726 | **28.06** | 66.77 |
>
> OUSAC achieves the highest CLIP Score (28.06) among all methods, surpassing even 50-step FLUX with full CFG. OUSAC w/o cache attains the best ImageReward (1.0092) while using only 8 CFG steps out of 20. Compared to TaylorSeer at similar computational cost (~216T vs ~205T MACs), OUSAC achieves  better quality (CLIP 28.06 vs 27.37, GenEval 66.77 vs 65.93).

---

> ### Author Response · Authors · 2025-11-27
>
> ## Weakness 2: Flexibility With the Learned Guidance Schedule
>
> We appreciate this thoughtful suggestion. We want to clarify that **our optimized schedules are NOT fully static**. They demonstrate significant flexibility through guidance scaling, which allows adaptation to different generation requirements without re-optimization. Specifically, OUSAC discovers an optimized schedule **w*** at a reference guidance scale, and we apply element-wise scaling k · **w*** to generalize to other guidance scales, where k is a multiplier.
>
> In the following, we present new experimental results validating this flexibility. Table R2 shows that on DiT-XL/2, a single optimized schedule transfers across guidance scales ×1 to ×5. Table R3 demonstrates the same property holds for FLUX, where the schedule generalizes from ×1 to ×3 while maintaining comparable quality to the baseline across all scales.
>
> **Table R2: DiT-XL/2 (ImageNet 256×256) with Different Guidance Scales**
>
> Reference guidance scale w=1.5. Multipliers: ×1 (w=1.5), ×3.3 (w≈5), ×5 (w=7.5).
>
>
> | Method | CFG-Steps ↓ | FID ↓ | sFID ↓ | Prec. ↑ | Recall ↑ |
> |--------|-----------|------|------|-------|--------|
> | DDIM (×1) | 50 | 2.23 | 4.29 | 80.06 | **59.20** |
>  OUSAC w/o cache (×1) | 8 | **2.10** |**4.29**| **81.85** | 58.80 |
>  | | | | | | | | | | |
> | DDIM (×3.3) | 50 | 16.39 | 15.53 | **92.22** | 22.76 |
> | OUSAC w/o cache (×3.3) | 8 | **9.20** | **9.23** | 90.31 | **39.60** |
> | | | | | | | | | | |
> | DDIM (×5) | 50 | 19.71 | 19.95 | **90.29** | 18.04 |
> | OUSAC w/o cache (×5) | 8 | **11.76** | **13.20** | 89.53 | **33.10** |
>
>
>
> **Table R3: FLUX (GenEval 512×512) with Different Guidance Scales**
>
> Reference guidance scale w=3.5. Multipliers: ×1 (w=1.5), ×2 (w=3), ×3 (w=4.5).
>
> | Method | Steps | CFG-Steps↓ | Position↑ | Colors↑ | Counting↑ | Color-Attr↑ | Two-Obj↑ | Single-Obj↑ | Overall↑ |
> |:--|:--:|:--:|:--:|:--:|:--:|:--:|:--:|:--:|:--:|
> | FLUX w/ True CFG (×1) | 50 | 50 | 19.50 | 80.32 | 77.19 | 49.75 | 86.11 | 98.75 | 68.60 |
> | OUSAC w/o cache (×1) | **20** | **8** | 21.25 | 80.32 | 68.44 | 51.50 | 84.85 | 98.44 | 67.46 |
> | | | | | | | | | | |
> | FLUX w/ True CFG (×2) | 50 | 50 | 23.75 | 73.67 | 75.31 | 40.25 | 82.32 | 95.94 | 65.20 |
> | OUSAC w/o cache (×2) | **20** | **8** | 20.50 | 79.52 | 65.31 | 36.75 | 79.04 | 98.75 | 63.31 |
> | | | | | | | | | | |
> | FLUX w/ True CFG (×3) | 50 | 50 | 20.75 | 53.72 | 62.81 | 21.25 | 72.22 | 90.31 | 53.51 |
> | OUSAC w/o cache (×3) | **20** | **8** | 16.50 | 72.34 | 56.88 | 23.25 | 61.87 | 96.25 | 54.51 |
>
>
>
> The results demonstrate that the discovered schedule transfers effectively across guidance scales without re-optimization. On DiT-XL/2, OUSAC consistently outperforms DDIM at all multipliers (×1, ×3.3, ×5) while using only 8 CFG steps instead of 50. On FLUX, OUSAC maintains competitive performance across ×1 (67.46 vs 68.60), ×2 (63.31 vs 65.20), and ×3 (54.51 vs 53.51). Both methods follow the same degradation trend at higher guidance scales, confirming that our schedule does not introduce additional sensitivity. This suggests that the relative importance of timesteps for guidance is scale-invariant.
>
> We also provide qualitative comparisons in the **Appendix of the revised manuscript**. **Figure 12** shows PixArt-α results where the optimized schedule w* (trained at w=4.5) is scaled by k=1, 1.5, 2 to generate images at different CFG strengths. **Figure 13** presents FLUX results with the same element-wise scaling approach (base schedule trained at w=1.5, scaled by ×1 and ×2). These visualizations confirm that k · w* maintains consistent visual quality across different CFG multipliers without re-optimization.
>
> ---
>
> **All results mentioned in this response have been added to the revised manuscript. The revised content are highlighted in blue.**
>
> ---

---

### Official Review · Reviewer_DXfQ · 2025-11-01

**Soundness:** 2
**Presentation:** 2
**Contribution:** 2
**Rating:** 4
**Confidence:** 5

**Summary:**

The paper proposes OUSAC, a two-stage, training-free framework to accelerate diffusion transformers (DiT). Stage-1 uses an evolutionary (gradient-free) search to discover per-timestep, sparse Classifier-Free Guidance (CFG) schedules so that CFG (which normally requires an unconditional and conditional forward pass per step) is applied only at a small subset of timesteps. Stage-2 compensates for the larger temporal feature shifts introduced by non-uniform guidance by performing adaptive rank allocation (regionwise SVD calibration) for cached transformer features via coordinate descent. On DiT-XL/2 (ImageNet) and PixArt-α (MSCOCO), OUSAC claims very large compute reductions (e.g., using guidance at 9/50 timesteps) with substantial MACs/latency savings and improved FID vs baselines (tables show up to ≈50–60% MAC savings while matching or improving FID). The paper also reports ablations showing adaptive rank allocation outperforms uniform calibration.

**Strengths:**

1. **Practical, training-free approach.** The method operates post-hoc on pretrained DiT models (no extra model training), which is attractive for adoption. The two stages are performed offline once per model and then used at inference.

2. **Diverse experiments.** Evaluation on large, realistic models/datasets (DiT-XL/2 on ImageNet 256/512; PixArt-α on MSCOCO) with standard metrics (FID, IS, sFID, CLIP score) and multiple baselines (ICC, L2C, Harmonica, DDIM) gives the paper empirical breadth. Reported improvements in both compute and FID are compelling in the tables.

3. **Ablations that probe components.** The paper includes ablations showing (a) the role of reference trajectory length in schedule discovery, and (b) the benefit of adaptive rank allocation vs a set of uniform ranks. These ablations increase trust that each component contributes.

**Weaknesses:**

- **Missing/insufficient reporting of optimization cost & reproducibility concerns.** The method’s Stage-1 uses evolutionary optimization (population sampling, multiple generations) to search a T-dimensional schedule; Stage-2 uses coordinate descent over region ranks. The paper gives search hyperparameters (e.g., 15 generations for DiT) but does not report the wall-clock compute, GPU hours, or search cost needed to discover schedules and rank assignments nor whether the search is practical for large models. This is crucial: a large offline search cost could negate the stated inference gains.

- **Ablation depth and hyperparameter sensitivity.** The method introduces multiple design choices and hyperparameters (e.g., threshold τ to disable unconditional pass, wmax, population size, number of generations, K regions for rank allocation, rmin/rmax). The paper gives some settings but lacks a systematic sensitivity analysis showing the method is not fragile to these choices.

- **Lack of relevant work.** The works TGate (which considers CFG and caching)[1], the classic cache acceleration method Delta-DiT[2], and the recent state-of-the-art cache methods TeaCache[3] and TaylorSeer[4] are currently not mentioned in the related work or included among the baselines.

- **Old pretrain model.** In this paper, we conduct experiments on two basic models, DiT-XL/2 and pixart-alpha, which are both models from a long time ago.


[1] Faster Diffusion via Temporal Attention Decomposition

[2] Δ-DiT: A Training-Free Acceleration Method Tailored for Diffusion Transformers

[3] Timestep Embedding Tells: It's Time to Cache for Video Diffusion Model

[4] From Reusing to Forecasting: Accelerating Diffusion Models with TaylorSeers

**Questions:**

1. **Robustness / transfer.** How does a schedule discovered for one guidance scale (e.g., baseline w=1.5) behave when used with different guidance scales, different prompts (e.g., long captions vs short), or different samplers? Can schedules transfer between model sizes (e.g., DiT-XL/2 → smaller DiT) or do they need to be rediscovered per model? Provide experiments or analysis.

2. **Hyperparameter sensitivity.** How sensitive are results to the threshold τ, wmax, number of generations, population size, and K (number of regions)? A short sweep or sensitivity table would strengthen claims of robustness.

3. **Interaction with other accelerations.** Can OUSAC be combined with sampling-reduction methods (DPM-Solver, progressive distillation) or pruning/quantization? If yes, does the guidance schedule discovery need adaptation? Experimental demonstration would increase the paper’s impact.

---

> ### Author Response · Authors · 2025-11-27
>
> We thank the reviewer for the constructive feedback. We address each concern below.
>
> ---
>
>
>
> ## W1: Optimization Cost & Reproducibility
>
>
> **Response:**
>
> We thank the reviewer for raising this important point. We provide detailed cost analysis for both optimization stages below.
>
> **Table R1: Stage-1 Evolutionary Search Cost Analysis** (4×H100 GPUs with parallel candidate evaluation)
>
> | | DiT-XL/2 | PixArt-α | FLUX |
> |---|----------|----------|------|
> | Search resolution | 256×256 | 128×128 | 128×128 |
> | Population size (P) | 16 | 32 | 32 |
> | Generations (G) | 10 | 15 | 15 |
> | Denoising steps (T) | 50 | 20 | 20 |
> | Reference steps (T_ref) | 1000 | 1000 | 100 |
> | Calibration prompts | 32 | 40 | 16 |
> | **Total evaluations** | 160 | 480 | 480 |
> | **Wall-clock time (hrs)** | 4.2 | 1.1 | 1.3 |
> | **GPU hours** | 16.8 | 4.4 | 5.3 |
>
> **Table R2: Stage-2 Adaptive Rank Allocation Cost Analysis** (4×H100 GPUs)
>
> | | DiT-XL/2 | PixArt-α | FLUX |
> |---|----------|----------|------|
> | Search resolution | 256×256 | 128×128 | 128×128 |
> | Number of regions (K) | 4 | 4 | 4 |
> | Rank range [r_min, r_max] | [16, 512] | [16, 256] | [16, 512] |
> | Denoising steps (T) | 50 | 20 | 20 |
> | Calibration images | 10,000 | 5,000 | 5,000 |
> | **Total FID evaluations** | 145 | 64 | 65 |
> | **Wall-clock time (hrs)** | 6.8 | 5.8 | 5.5 |
> | **GPU hours** | 27.2 | 23.2 | 22.0 |
>
>
> 1. While PixArt-α and FLUX are trained at 256×256 and 1024×1024 resolution respectively, we conduct evolutionary search at smaller resolutions (128×128) to significantly reduce search time and computational cost. Interestingly, we found that ***the discovered schedules from low resolution (128×128) transfer well to high resolution (1024×1024).*** This shows that our approach can be used efficiently in practice. We have updated the paper accordingly and provide visualization results in the Appendix: Figure 10 shows FLUX generation with short and long prompts comparing our method against the baseline and TaylorSeer [1], while Figure 11 presents PixArt-α results comparing our method with the baseline.
>
>
> 2. **One-time offline cost:** The optimization is performed *once per model architecture* and the discovered schedules generalize across different prompts, seeds, and conditions.
>
> ---
>
> ## W2 & Q2: Hyperparameter Sensitivity
>
>
> **Response:**
>
> We appreciate this feedback and provide sensitivity analysis for the key hyperparameters below.
>
> ### Sensitivity to K (Number of Regions)
>
> We analyze the impact of region count K on adaptive rank allocation for DiT-XL/2 (ImageNet 256×256). As shown in Figure 8 (Appendix, revised manuscript):
>
> | K | 1 | 4 | 7 | 14 |
> |---|---|---|---|---|
> | FID | 2.12 | 2.05 | 2.03 | 2.04 |
>
> While K=7 achieves the lowest FID (2.03), K=4 offers a favorable trade-off between quality (FID 2.05) and optimization efficiency, requiring significantly fewer coordinate descent evaluations due to the reduced search space. The results demonstrate that OUSAC is robust to the choice of K—performance remains stable across a wide range (K=4 to K=14), with only 0.02 FID difference.
>
>
> ### Sensitivity to Population Size
> We analyze the impact of population size on DiT-XL/2 (ImageNet 256×256):
>
> | Population Size | 16 | 24 | 32 |
> |---|---|---|---|
> | FID | 2.10 | 2.13 | 2.09 |
>
> The results show that OUSAC is robust to population size, with only 0.04 FID difference across the range. A population size of 16 is sufficient, as larger populations provide no significant improvement. We use 16 in our experiments for computational efficiency.
>
>
> ### Sensitivity to Number of Generations (G)
>
> We provide convergence analysis of the evolutionary optimization for PixArt-α in Figure 7 (Appendix, revised manuscript):
>
> - **Fitness convergence:** Best fitness improves rapidly in early generations and plateaus around generation 20, indicating stable convergence.
> - **MSE convergence:** The MSE between optimized and reference generations stabilizes at ~0.16–0.18, reflecting the trade-off between quality preservation and sparsity.
>
> In practice, we use G=15 generations, which achieves near-optimal schedules while reducing optimization overhead. The convergence curves show that even G=10 would yield competitive results, demonstrating robustness to this hyperparameter.
>
>
> ### Sensitivity to w_max
> We analyze the impact of w_max on PixArt-α with base guidance scale w=4.5:
>
> | w_max | 5 | 10 | 12 | 15 |
> |---|---|---|---|---|
> | FID | 23.60 | 22.69 | 22.85 | 23.01 |
>
> The results show that performance stabilizes when w_max ≥ 10, with only 0.32 FID difference across the range (10–15). We use w_max=10 in our experiments as it achieves the best FID while providing sufficient search space for the evolutionary optimization.

---

> ### Author Response · Authors · 2025-11-27
>
> ## W3: Related Work
>
>
> **Response:**
>
> We thank the reviewer for this suggestion. We have included TGate [2], Delta-DiT [3], TeaCache [4], and TaylorSeer [1] in the Related Work section of our revised manuscript.
>
> For experimental comparisons, we have included TaylorSeer [1] as a baseline on DiT-XL/2 512×512 and FLUX. Detailed results and analysis are provided in our response to W4 & Q1 below.
>
> ---
>
> ## W4 & Q1: Extension to FLUX & Robustness Across Guidance Scales
>
> **Response:**
>
> We thank the reviewer for these important questions. We address both concerns below.
>
> ### Results on Modern Models (FLUX)
>
> We have extended our experiments to FLUX, a state-of-the-art flow-based model. Although FLUX was trained with CFG distillation, recent developments enable explicit CFG: the official support for a "True CFG" mode and CFG-Zero* [5] (whose Table 3 demonstrates that explicit CFG enhances FLUX's output quality). We evaluate OUSAC under the official True CFG setting on DrawBench and GenEval:
>
> **Table R3: DrawBench evaluation & GenEval evaluation with FLUX at 512×512 resolution.**
>
> | Method | Steps | CFG-Steps↓ | MACs(T)↓ | Latency(s)↓ | DrawBench ImageReward↑ | DrawBench CLIP↑ | GenEval  Overall↑ |
> |--------|-------|-----------|----------|----------|--------------|-------|-------------------|
> | FLUX (True CFG=1.5) | 50 | 50 | 1143.82 | 52.01 |  0.9956 | 27.70 | 68.60 |
> | FLUX (True CFG=1.5) | 20 | 20 | 457.52 | 21.24 |  0.8950 | 27.55 | 66.05 |
> | FLUX (True CFG=1.5) | 16 | 16 | 366.02 | 17.19 |  0.7866 | 27.46 | 63.59 |
> | FLUX | 50 | 0 | 571.48 | 26.43 |   0.9296 | 27.23 | 65.38 |
> | FLUX | 20 | 0 | 228.76 | 10.87 |   0.8934 | 27.03 | 65.26 |
> | TaylorSeer (N=3, O=2) | 50 | 50 | 411.77 | 28.30 |  1.0022 | 27.78 | 67.70 |
> | TaylorSeer (N=6, O=2) | 50 | 50 | 228.76 | 20.10 |   0.8492 | 27.26 | 59.58 |
> | TaylorSeer (N=3, O=2) | 50 | 0 | 205.88 | 13.87 |   0.9445 | 27.37 | 65.93 |
> | **OUSAC w/o cache** | 20 | 8 | 320.26 | 15.15 |   **1.0092** | **28.06** | **67.46** |
> | **OUSAC** | 20 | 8 | 216.48 | 14.88 |   0.9726 | **28.06** | 66.77 |
>
> OUSAC achieves the highest CLIP Score (28.06) among all methods, surpassing even 50-step FLUX with full CFG. OUSAC w/o cache attains the best ImageReward (1.0092) while using only 8 CFG steps out of 20. Compared to TaylorSeer at similar computational cost (~216T vs ~205T MACs), OUSAC achieves  better quality (CLIP 28.06 vs 27.37, GenEval 66.77 vs 65.93).

---

> ### Author Response · Authors · 2025-11-27
>
> ### Robustness to Different Guidance Scales
>
> OUSAC discovers an optimized schedule **w*** at a reference guidance scale. To evaluate whether this schedule generalizes to other guidance scales, we apply element-wise scaling: k · **w***, where k is a multiplier.
>
> **Table R4: DiT-XL/2 (ImageNet 256×256) with Different Guidance Scales**
>
> Reference guidance scale w=1.5. Multipliers: ×1 (w=1.5), ×3.3 (w≈5), ×5 (w=7.5).
>
> | Method | CFG-Steps ↓ | FID ↓ | sFID ↓ | Prec. ↑ | Recall ↑ |
> |--------|-----------|------|------|-------|--------|
> | DDIM (×1) | 50 | 2.23 | 4.29 | 80.06 | **59.20** |
>  OUSAC w/o cache (×1) | 8 | **2.10** |**4.29**| **81.85** | 58.80 |
>  | | | | | | | | | | |
> | DDIM (×3.3) | 50 | 16.39 | 15.53 | **92.22** | 22.76 |
> | OUSAC w/o cache (×3.3) | 8 | **9.20** | **9.23** | 90.31 | **39.60** |
> | | | | | | | | | | |
> | DDIM (×5) | 50 | 19.71 | 19.95 | **90.29** | 18.04 |
> | OUSAC w/o cache (×5) | 8 | **11.76** | **13.20** | 89.53 | **33.10** |
>
> **Table R5: FLUX (GenEval 512×512) with Different Guidance Scales**
>
> Reference guidance scale w=3.5. Multipliers: ×1 (w=1.5), ×2 (w=3), ×3 (w=4.5).
>
> | Method | Steps | CFG-Steps↓ | Position↑ | Colors↑ | Counting↑ | Color-Attr↑ | Two-Obj↑ | Single-Obj↑ | Overall↑ |
> |:--|:--:|:--:|:--:|:--:|:--:|:--:|:--:|:--:|:--:|
> | FLUX w/ True CFG (×1) | 50 | 50 | 19.50 | 80.32 | 77.19 | 49.75 | 86.11 | 98.75 | 68.60 |
> | OUSAC w/o cache (×1) | **20** | **8** | 21.25 | 80.32 | 68.44 | 51.50 | 84.85 | 98.44 | 67.46 |
> | | | | | | | | | | |
> | FLUX w/ True CFG (×2) | 50 | 50 | 23.75 | 73.67 | 75.31 | 40.25 | 82.32 | 95.94 | 65.20 |
> | OUSAC w/o cache (×2) | **20** | **8** | 20.50 | 79.52 | 65.31 | 36.75 | 79.04 | 98.75 | 63.31 |
> | | | | | | | | | | |
> | FLUX w/ True CFG (×3) | 50 | 50 | 20.75 | 53.72 | 62.81 | 21.25 | 72.22 | 90.31 | 53.51 |
> | OUSAC w/o cache (×3) | **20** | **8** | 16.50 | 72.34 | 56.88 | 23.25 | 61.87 | 96.25 | 54.51 |
>
>
> The results demonstrate that the discovered schedule transfers effectively across guidance scales **without re-optimization**. On DiT-XL/2, OUSAC consistently outperforms DDIM at all multipliers (×1, ×3.3, ×5) while using only 8 CFG steps instead of 50. On FLUX, OUSAC maintains competitive performance across ×1 (67.46 vs 68.60), ×2 (63.31 vs 65.20), and ×3 (54.51 vs 53.51). Both methods follow the same degradation trend at higher guidance scales, confirming that our schedule does not introduce additional sensitivity. This suggests that the relative importance of timesteps for guidance is scale-invariant.
>
>
> We also provide qualitative comparisons in the **Appendix of the revised manuscript**. **Figure 12** shows PixArt-α results where the optimized schedule w* (trained at w=4.5) is scaled by k=1, 1.5, 2 to generate images at different CFG strengths. **Figure 13** presents FLUX results with the same element-wise scaling approach (base schedule trained at w=1.5, scaled by ×1 and ×2). These visualizations confirm that k · w* maintains consistent visual quality across different CFG multipliers without re-optimization.
>
> ### Robustness to Different Prompt Lengths
>
>
> We have updated the paper and provide visualization results in the **Appendix of the revised manuscript**: **Figure 10** shows FLUX generation with **short prompts** and **long prompts** comparing our method against the baseline and TaylorSeer [1], while **Figure 11** presents PixArt-α results comparing our method with the baseline. These visualizations cover diverse categories including objects, scenes, and complex compositions, demonstrating that OUSAC maintains consistent visual quality across varying prompt lengths and semantic complexity while achieving significant computational savings.
>
>
>
> ### Transfer Across Models
>
> We appreciate the suggestion by the reviewer to test on other DiT models.
> Unfortunately, to the best of our knowledge, DiT-XL/2 is the only publicly available pretrained checkpoint in the DiT model family ([DiT Official PyTorch Implementation](https://github.com/facebookresearch/DiT?tab=readme-ov-file#sampling--)).
>
> It is computationally prohibitive to train our own DiT model from scratch within the ICLR discussion period.
>
> We will be happy to conduct such transfer experiments and include the results if the reviewer could kindly give us pointers to publicly available pretrained checkpoint for such DiT models.

---

> ### Author Response · Authors · 2025-11-27
>
> ---
>
> ## Q3: Compatibility with Other Acceleration Methods
>
> > "Can OUSAC be combined with sampling-reduction methods (DPM-Solver, progressive distillation) or pruning/quantization?"
>
> **Response:**
>
> ### Combination with DPM-Solver
>
> Yes, our PixArt-α experiments already use DPM-Solver as the base sampler, demonstrating that OUSAC combines effectively with advanced sampling methods without modification. Please see Table 2 in the mian paper.
>
> ### Combination with Quantization (INT8)
>
> Thank you for the suggestion. We have conducted new experiment of combining OUSAC with quantitzation. We happy to report that OUSAC can be directly applied with INT8 quantization without modification. As shown below, OUSAC consistently outperforms DDIM under both FP16 and INT8 precision:
>
>
>
> | Precision | Method | CFG-Steps↓ | IS↑ | FID↓ | sFID↓ |
> |-----------|--------|-----------|-----|-----|------|
> | FP16 (original) | DDIM | 50 | 239.4 | 2.23 | 4.29 |
> | FP16 (original) | OUSAC w/o cache | 8 | **263.0** | **2.10** | **4.29** |
> | ─────── | ─────── | ─────── | ─── | ─── | ─── |
> | INT8 (quantization) | DDIM | 50 | 199.9 | 4.61 | 9.17 |
> | INT8 (quantization) | OUSAC w/o cache | 8 | **225.4** | **3.61** | **8.55** |
>
> **Please note that we don't need to run any new training on the INT8 model**. This table shows that the discovered sparse guidance schedules from the original precision transfer well directly to quantized models, demonstrating that OUSAC can be combined with quantization techniques.
>
> We will include this new experiment in the revision of our paper.
>
>
> ---
>
> **All results mentioned in this response have been added to the revised manuscript. The revised content are highlighted in blue.**
>
> ---
>
>
> [1] Liu, Jiacheng, et al. "From reusing to forecasting: Accelerating diffusion models with taylorseers." (2025).
>
> [2] Liu, Haozhe, et al. "Faster diffusion via temporal attention decomposition." arXiv preprint arXiv:2404.02747 (2024).
>
> [3] Chen, Pengtao, et al. "△-DiT: A Training-Free Acceleration Method Tailored for Diffusion Transformers." arXiv preprint arXiv:2406.01125 (2024).
>
> [4] Liu, Feng, et al. "Timestep Embedding Tells: It's Time to Cache for Video Diffusion Model." Proceedings of the Computer Vision and Pattern Recognition Conference. 2025.
>
> [5] Fan, Weichen, et al. "Cfg-zero*: Improved classifier-free guidance for flow matching models."  (2025).
>
>
> We hope our response addresses the reviewer's concerns and are happy to provide further clarification.

---

### Official Review · Reviewer_VFpQ · 2025-11-01

**Soundness:** 3
**Presentation:** 3
**Contribution:** 2
**Rating:** 4
**Confidence:** 5

**Summary:**

The paper introduces OUSAC, a two-stage optimization framework to accelerate Diffusion Transformers (DiT) under Classifier-Free Guidance (CFG). Stage 1 uses evolutionary search to learn sparse, timestep-specific guidance schedules that skip unconditional passes when guidance is unimportant. Stage 2 adds adaptive rank allocation for increment-calibrated caching, assigning different SVD ranks to transformer regions to handle the feature inconsistencies caused by variable guidance. Experiments on DiT-XL/2 (ImageNet 512×512) and PixArt-α (MSCOCO 256×256) report up to 53–60 % computation reduction with modest or improved FID (e.g., 2.72 vs 3.20 on DiT-XL/2). The authors claim that OUSAC is training-free, improves both efficiency and quality, and generalizes across prompts.

**Strengths:**

1. The paper is clearly written and the motivation is easy to follow.

2. Integrating guidance scheduling and caching is a practical and coherent idea.

3. Results on DiT and PixArt-α show consistent speedup with comparable quality.

4. The approach is training-free and could be added to existing DiT models.

**Weaknesses:**

1. Limited relevance: The method assumes Classifier-Free Guidance, but many recent models (e.g., FLUX) do not use CFG at all. The paper does not explain how OUSAC would work in those settings.

2. Missing search-cost details: Stage-1 evolutionary search likely requires many full generations, but the paper gives no runtime, GPU hours, or total evaluations.

3. Generalization claims unproven: The paper claims that the discovered guidance schedule “generalizes across different prompts and conditions,” but provides no explanation or evidence for why this should hold. In practice, the optimal guidance steps are likely input-dependent, since noise trajectories and attention dynamics vary with prompt complexity. The current approach learns a single fixed sparse pattern, which seems more dataset-specific than truly generalizable.

4. Incremental novelty: Both sparse CFG and caching are known ideas; OUSAC combines them but adds limited new insight.

5. Small improvements: The reported FID gains (≈0.2–0.3) are minor and within normal variance.

**Questions:**

1. How does OUSAC apply to CFG-free models like FLUX?

2. What is the total cost of the evolutionary search (generations, population size, GPU hours)?

3. Why should the discovered schedule generalize to new prompts or datasets?

4. What is the actual cache-reuse rate in practice?

---

> ### Author Response · Authors · 2025-11-27
>
> We thank the reviewer for the constructive feedback. We address each concern below.
>
> ---
>
> ## W1/Q1: How does OUSAC apply to CFG-free models like FLUX?
>
> We thank the reviewer for this important question. We respectfully clarify that **CFG remains highly relevant and even essential for modern flow-based models**, such as Flux and SD 3.5, based on the following evidence:
>
> 1. **SD 3.5 employs CFG and achieves leading performance**.  Stable Diffusion 3.5 uses CFG for generation. According to Stability AI's official analysis, SD 3.5 Large "leads the market in prompt adherence and rivals much larger models in image quality" [1], demonstrating that CFG-based models remain at the forefront of text-to-image generation.
>
> 2. **FLUX officially supports CFG.** Both the official FLUX repository and HuggingFace Diffusers implement "True CFG" mode, which performs separate conditional and unconditional forward passes—exactly the setting our method targets.
>
> 3. **CFG improves generation quality on FLUX.** As shown in Table 3 of CFG-Zero* [2], CFG enhances FLUX's output quality, with FLUX using True CFG achieving higher Aesthetic Score and CLIP Score than SD 3.5.
>
> We apply OUSAC to FLUX with True CFG, and compare it with FLUX without CFG, FLUX with True CFG, and a state-of-the-art cache-only method TaylorSeer [3] for acceleration. We use DrawBench and GenEval for evaluation and the results are in the table below:
>
> **Table R1: DrawBench evaluation & GenEval evaluation with FLUX at 512×512 resolution.**
>
> | Method | Steps | CFG-Steps↓ | MACs(T)↓ | Latency(s)↓ | DrawBench ImageReward↑ | DrawBench CLIP↑ | GenEval  Overall↑ |
> |--------|-------|-----------|----------|----------|--------------|-------|-------------------|
> | FLUX (True CFG=1.5) | 50 | 50 | 1143.82 | 52.01 |  0.9956 | 27.70 | 68.60 |
> | FLUX (True CFG=1.5) | 20 | 20 | 457.52 | 21.24 |  0.8950 | 27.55 | 66.05 |
> | FLUX (True CFG=1.5) | 16 | 16 | 366.02 | 17.19 |  0.7866 | 27.46 | 63.59 |
> | FLUX | 50 | 0 | 571.48 | 26.43 |   0.9296 | 27.23 | 65.38 |
> | FLUX | 20 | 0 | 228.76 | 10.87 |   0.8934 | 27.03 | 65.26 |
> | TaylorSeer (N=3, O=2) | 50 | 50 | 411.77 | 28.30 |  1.0022 | 27.78 | 67.70 |
> | TaylorSeer (N=6, O=2) | 50 | 50 | 228.76 | 20.10 |   0.8492 | 27.26 | 59.58 |
> | TaylorSeer (N=3, O=2) | 50 | 0 | 205.88 | 13.87 |   0.9445 | 27.37 | 65.93 |
> | **OUSAC w/o cache** | 20 | 8 | 320.26 | 15.15 |   **1.0092** | **28.06** | **67.46** |
> | **OUSAC** | 20 | 8 | 216.48 | 14.88 |   0.9726 | **28.06** | 66.77 |
>
> OUSAC achieves the highest CLIP Score (28.06) among all methods, surpassing even 50-step FLUX with full CFG. OUSAC w/o cache attains the best ImageReward (1.0092) while using only 8 CFG steps out of 20. Compared to TaylorSeer at similar computational cost (~216T vs ~205T MACs), OUSAC achieves  better quality (CLIP 28.06 vs 27.37, GenEval 66.77 vs 65.93).
>
>
>
>
> ## W2/Q2: Search-Cost Details
>
> Thank you for raising this important point. We provide the complete Stage-1 evolutionary search cost analysis below:
>
> **Table R2: Stage-1 Evolutionary Search Cost Analysis**
>
> | | **DiT-XL/2** | **PixArt-α** | **FLUX** |
> |:--|:--:|:--:|:--:|
> | Search resolution | 256×256 | 128×128 | 128×128 |
> | Population size (*P*) | 16 | 32 | 32 |
> | Generations (*G*) | 10 | 15 | 15 |
> | Denoising steps (*T*) | 50 | 20 | 20 |
> | Reference steps (*T*_ref) | 1000 | 1000 | 100 |
> | Calibration prompts | 32 | 40 | 16 |
> | **Total evaluations** | 160 | 480 | 480 |
> | **Wall-clock time (hrs)** | 4.2 | 1.1 | 1.3 |
> | **GPU hours** | 16.8 | 4.4 | 5.3 |
>
> All experiments use 4×H100 GPUs with parallel candidate evaluation via multiprocessing.
>
> While PixArt-α and FLUX are trained at 256×256 and 1024×1024 resolution respectively, we conduct evolutionary search at smaller resolutions (128×128) to significantly reduce search time and computational cost. Interestingly, we found that ***the discovered schedules from low resolution (128×128) transfer well to high resolution (1024×1024)***. We have updated the main paper and provide visualization results in the Appendix: **Figure 10** shows FLUX generation with short and long prompts comparing our method against the baseline and TaylorSeer [3], while **Figure 11** presents PixArt-α results comparing our method with the baseline. This shows that our approach can be used efficiently in practice.

---

> ### Author Response · Authors · 2025-11-27
>
> ## W3/Q3: Generalization
>
>
> Thank you for this important question. We provide both quantitative and qualitative evidence demonstrating that discovered schedules generalize well across different prompts and datasets:
>
> ### Cross-Dataset Evaluation
>
> 1. **PixArt-α**: We add evaluation on **PartiPrompts** [4], a separate benchmark with diverse compositional prompts. As shown in the Table R3 below, OUSAC achieves the best PartiPrompts score (17.92) while maintaining strong MSCOCO performance, demonstrating generalization across datasets with different prompt distributions.
>
> **Table R3:**
> | Method | Steps | CFG-Steps↓ | MACs (T)↓ | FID↓ | MSCOCO CLIP Score↑ | PartiPrompts CLIP Score↑ |
> |:--|:--:|:--:|:--:|:--:|:--:|:--:|
> | DPM-Solver | 1000 | 1000 | 336.11 | 22.97 | 16.42 | 17.31 |
> | DPM-Solver | 20 | 20 | 6.72 | 24.60 | 16.31 | 17.36 |
> | ICC | 20 | 20 | 3.70 | 21.86 | 16.47 | 17.20 |
> | **OUSAC w/o cache** | 20 | **6** | 4.37 | 22.69 | 16.39 | **17.92** |
> | **OUSAC** | 20 | **6** | **2.67** | **19.27** | **16.48** | 17.45 |
>
> 2. **FLUX**: The evolutionary search is conducted using prompts sampled from **COCO captions 30k** [5], but evaluation is performed on completely different benchmarks—**GenEval** [6] and **DrawBench** [7]. As shown in Table R1, OUSAC achieves 5.28× speedup (1143.82T → 216.48T MACs) compared to 50-step FLUX with True CFG, while only incurring minor quality degradation (ImageReward: 0.9956 → 0.9726, GenEval: 68.60 → 66.77). This cross-dataset transfer provides strong evidence for generalization.
>
> ### Qualitative Visualization
> We have updated the main paper and provide visualization results in the Appendix: Figure 10 shows FLUX generation with **short prompts** and **long prompts** comparing our method against the baseline and TaylorSeer [3], while Figure 11 presents PixArt-α results comparing our method with the baseline. These visualizations cover diverse categories including objects, scenes, and complex compositions, demonstrating that OUSAC maintains consistent visual quality across varying prompt lengths and semantic complexity while achieving significant computational savings.

---

> ### Author Response · Authors · 2025-11-27
>
> ## W4: Novelty and Contributions
>
> We thank the reviewer for raising this question. We respectfully clarify that OUSAC is **fundamentally different** from existing sparse CFG methods.
>
> **Existing methods** (Adaptive Guidance [14], DiTFastAttn [15], FasterCache [16]) decide when to skip computation based on cond/uncond output similarity:
> - The **total sampling steps** remain **unchanged**; only unconditional or conditional forward passes can be reduced (e.g., 50 steps with 50 cond + 25 uncond ≈ quality of 50 steps with 50 cond + 50 uncond)
> - The guidance scale remains **fixed** when CFG is applied
>
> **In contrast, OUSAC** uses evolutionary optimization to jointly search sparse patterns and guidance scales:
> - The **total sampling steps** can be significantly **reduced** (e.g., 20 steps with 20 cond + 8 uncond ≈ quality of 50 steps with 50 cond + 50 uncond)
> - The guidance scale is **variable** across timesteps
>
> Moreover, we discover that optimized schedules exhibit **cross-scale generalization**: a single optimized schedule w* transfers to different CFG strengths by simply multiplying a scalar k (i.e.,  k · w*), eliminating per-scale re-optimization. Beyond guidance optimization (Stage 1), OUSAC is the **first** to introduce **feature caching** (Stage 2) into CFG scheduling for additional acceleration. We elaborate below:
>
> **(a.) Why step reduction matters beyond MACs savings.**
>
> Feature caching methods like TaylorSeer [3] and ICC [8] achieve significant MACs reduction but face challenges in translating this to real-world latency improvements. On DiT-XL/2 256×256, a user reported only 0.1s latency reduction (0.8s → 0.69s) despite 3.57× MACs savings (GitHub Issue [#22](https://github.com/Shenyi-Z/TaylorSeer/issues/22)). The authors explained:
>
> > *"The reason for this is that the computation during DiT inference is very sparse... due to the high sparsity of DiT's computation, the speedup ratio may vary significantly across different devices."*
>
> To reproduce their reported latency, they recommend *"inference with a large batch size"*, a setting that does not reflect typical single-image generation (batch size = 1).
>
> In contrast, **sparse CFG and step reduction directly eliminate forward passes**, providing guaranteed **latency reduction** regardless of batch size or hardware configuration. Our experiments (Table 8 in the revised manuscript) confirm this: when generating 8 images sequentially with batch size 1, OUSAC achieves 14.88s latency vs. TaylorSeer's 20.10s at comparable MACs (216.48T vs. 228.76T).
>
> **(b.) Why evolutionary optimization is necessary.**
>
> OUSAC reduces forward passes by optimizing **variable guidance scales** across timesteps. This is inspired by recent CFG scheduling research (E-CFG [10], β-CFG [11], RAAG [12], FDG [13]) that shows time-varying guidance can improve generation quality. However, these methods still require **full CFG computation** at every timestep, and as E-CFG [10] observes, their parametric schedules *"are still largely **heuristic** in nature."*
>
> Extending such heuristic schedules to also reduce computation (i.e., allowing some timesteps to skip CFG entirely) creates a **joint discrete-continuous search space** where manual design becomes **intractable**. Instead, OUSAC uses **evolutionary optimization** to automatically discover sparse patterns and guidance scales in a unified framework, solving an optimization problem that is infeasible for manual heuristics.
>
> **(c.) Feature caching strategy.**
>
> OUSAC is the first to integrate guidance scheduling with adaptive caching. We derive closed-form expressions for how variable guidance affects the denoising trajectory:
>
> - **Guidance scale variation** (Eq. 17): $\Delta x_{t-1}^{\text{scale}} = \beta_{t-1,t}(w_{t-1}^* - w_t^*)[\epsilon_c(x_t, t) - \epsilon_u(x_t, t)]$
> - **Branch switching** (Eq. 18): $\Delta x_{t-1}^{\text{switch}} = \beta_{t-1,t}(1 - w_t^*)[\epsilon_c(x_t, t) - \epsilon_u(x_t, t)]$
>
> These derivations explain *why* variable guidance breaks standard caching assumptions. Figure 6 (Appendix) confirms that variable guidance causes 40% higher final latent error, and Figures 7–8 demonstrate heterogeneous error distribution across transformer blocks. This validates the need for **adaptive rank allocation** (not uniform) to compensate for irregular feature deviations.

---

> ### Author Response · Authors · 2025-11-27
>
> **(d.) Summary of contributions.**
>
> - **First hybrid discrete-continuous guidance optimization**: jointly optimizes *when to skip* (discrete) and *what scale to use* (continuous)
> - **Evolutionary optimization for large-scale models**: only requires forward passes, enabling optimization on 12B-parameter FLUX where gradient-based NAS is infeasible
> - **First integration of guidance scheduling with adaptive caching**: addresses the unexplored variable-guidance caching incompatibility
> - **Cross-scale generalization via $k·w^*$**: existing parametric schedules [9,10,11,12,13] require re-tuning (as noted in [11]: *"optimal parameters do not generalize"*); OUSAC transfers via simple multiplicative scaling (see **Table 11, Figures 12–13 in Appendix**)
>
> Across three architectures (DiT-XL/2, PixArt-α, FLUX), OUSAC consistently achieves 60–70% computational savings while maintaining or improving generation quality.
>
> In summary, OUSAC is **not just a combination** of sparse CFG and caching. Our key insight is that **variable guidance is essential** to skip computation steps without losing quality. By solving this optimization problem, OUSAC turns theoretical savings into **guaranteed real-world speedups**.
>
> ---
>
> ## W5: Regarding FID Improvements
>
> We respectfully clarify that **the main focus of OUSAC is computational efficiency, not FID improvement**. While efficiency-driven acceleration methods are typically expected to match or slightly underperform the original model, we are pleasantly surprised that OUSAC can achieve FID gains. Our main contribution is achieving significant computation reduction while matching state-of-the-art quality.
>
>
>
> ## Q4: Cache-Reuse Rate
> We use a fixed caching configuration: consecutive CFG steps require full computation, while non-CFG regions use N=2 or N=3 caching (reuse 1–2 steps per computation). This achieves an overall **50% reuse ratio** across the discovered guidance pattern. Notably, our method focuses primarily on CFG scheduling, so cache reuse is not the main focus of our work.
>
>
>
> ---
>
> **All results mentioned in this response have been added to the revised manuscript. The revised content are highlighted in blue.**
>
> ---
>
> [1] Stability AI. "Introducing Stable Diffusion 3.5." (2024).
>
> [2] Fan, Weichen, et al. "Cfg-zero*: Improved classifier-free guidance for flow matching models."  (2025).
>
> [3] Liu, Jiacheng, et al. "From reusing to forecasting: Accelerating diffusion models with taylorseers." (2025).
>
> [4] Yu, Jiahui, et al. "Scaling autoregressive models for content-rich text-to-image generation."  (2022).
>
> [5] Lin, Tsung-Yi, et al. "Microsoft coco: Common objects in context." ECCV 2014.
>
> [6] Ghosh, Dhruba, Hannaneh Hajishirzi, and Ludwig Schmidt. "Geneval: An object-focused framework for evaluating text-to-image alignment." NIPS (2023).
>
> [7] Saharia, Chitwan, et al. "Photorealistic text-to-image diffusion models with deep language understanding." NIPS (2022).
>
> [8] Chen, Zhiyuan, et al. "Accelerating Diffusion Transformer via Increment-Calibrated Caching with Channel-Aware Singular Value Decomposition." CVPR. 2025.
>
> [9] Kynkäänniemi, Tuomas, et al. "Applying guidance in a limited interval improves sample and distribution quality in diffusion models." NIPS (2024).
>
> [10] Gao, Jiayang, et al. "Beyond Fixed: Aligning Guidance with Diffusion Dynamics via Exponential Scaling."
>
> [11] Malarz, Dawid, et al. "Classifier-free guidance with adaptive scaling." (2025).
>
> [12] Zhu, Shangwen, et al. "RAAG: Ratio Aware Adaptive Guidance." arXiv preprint arXiv:2508.03442 (2025).
>
> [13] Sadat, Seyedmorteza, et al. "Guidance in the Frequency Domain Enables High-Fidelity Sampling at Low CFG Scales." arXiv preprint arXiv:2506.19713 (2025).
>
> [14] Castillo, Angela, et al. "Adaptive guidance: Training-free acceleration of conditional diffusion models." AAAI 2025.
>
> [15] Yuan, Zhihang, et al. "Ditfastattn: Attention compression for diffusion transformer models." NIPS (2024).
>
> [16] Lv, Zhengyao, et al. "Fastercache: Training-free video diffusion model acceleration with high quality." (2024).
>
> [17] Dinh, Anh-Dung, Daochang Liu, and Chang Xu. "Compress Guidance in Conditional Diffusion Sampling." arXiv preprint arXiv:2408.11194 (2024).
>
> [18] Wang, Xi, et al. "Analysis of classifier-free guidance weight schedulers." arXiv preprint arXiv:2404.13040 (2024).

---

### Comment · Area_Chair_gjCk · 2025-11-28

Dear reviewers,

Please check authors' responses and provide your feedback.

AC

---

### Author Response · Authors · 2025-12-03

## Authors' Summary

We thank all reviewers for their constructive feedback and suggestions. During the discussion period, we conducted additional quantitative experiments on new datasets and benchmarks, compared against new SOTA caching method, tested on modern flow-based method (FLUX), and performed more ablation analysis. **Our method OUSAC has demonstrated consistent superior performance in these new results**. We also carefully consult the literature and provides new insights about our contribution comparing with other acceleration or caching methods.



We are glad that reviewers praise our approach being **practical and training-free** [VFpQ, DXfQ, ofxV], **clearly written** with **easy-to-follow** motivation [VFpQ], featuring **diverse experiments** and **thorough ablations** [DXfQ, ofxV], and being **the first to recognize** the interdependence between variable guidance and cache calibration with a **particularly original** and **technically solid** optimization framework [ofxV].

We summarize main concerns from the reviewers and our responses below. We appreciate the reviewers to help us make our paper better and stronger.


---

### 1. Evaluation on SOTA T2I Models and Benchmarks [VFpQ, DXfQ, ofxV, fE4p]

**Concerns:** Reviewers requested additional evaluations on state-of-the-art flow-based models like FLUX and comprehensive T2I benchmarks with compositional and semantic metrics (i.e., CLIP Score, GenEval, ImageReward).

**Response:** We confirm that the OUSAC works well on FLUX and report the results on benchamrks DrawBench and GenEval. OUSAC achieves the highest CLIP Score (28.06) and ImageReward (1.0092), surpassing 50-step FLUX with full CFG while using only 8 CFG steps out of 20. At comparable MACs (~216T), OUSAC outperforms TaylorSeer, a SOTA caching method, in quality (CLIP 28.06 vs 27.37, GenEval 66.77 vs 65.93).

---

### 2. Generalization [VFpQ, DXfQ, ofxV]

**Concerns:** Reviewers asked if our discovered CFG schedules generalize to new prompts and different CFG scales.

**Response:**
We confirm that OUSAC generalizes well in both scenarios.

*(a) Generalization across Prompts*: Schedules discovered on COCO captions transfer effectively to different benchmarks. On FLUX, testing OUSAC on DrawBench and GenEval achives 5.28× speedup with minimal quality loss. On PixArt-α, OUSAC achieves the best PartiPrompts score (17.92) while maintaining strong MSCOCO performance.


*(b) Generalization across CFG Scales:* Via simple multiplicative scaling k·w*, we can transfer schedules across guidance scales without re-optimization. On DiT-XL/2, OUSAC outperforms DDIM at all multipliers (×1, ×3.3, ×5) while using only 8 CFG steps. On FLUX, OUSAC maintains competitive performance from ×1 to ×3.

---

### 3. Optimization Cost [VFpQ, DXfQ, fE4p]

**Concerns**: Reviewers requested the optimization cost details and asked about OUSAC's scalability to large models.

**Response:** For FLUX (12B parameters), total optimization takes ~7 hours on 4×H100 GPUs (Stage-1: ~1.3 hrs, Stage-2: ~5.5 hrs). To reduce search cost, we conduct evolutionary search at 128×128 resolution; the discovered schedules **transfer directly to 1024×1024 inference without re-optimization**, as validated by visualization results (Figures 10–11 in Appendix). This is a **one-time offline cost** per architecture that generalizes across prompts and conditions.


---


### 4. Novelty [VFpQ, fE4p]

**Concerns**: Reviewers argued that OUSAC simply combines existing techniques (sparse CFG + caching).

**Response:** OUSAC is *fundamentally different* from existing CFG-related methods, which can be categorized into:

- [Sparse CFG methods] (i.e., Adaptive Guidance, DiTFastAttn, FasterCache): they reduce computation but *sacrafice quality* with fixed guidance scales;
- [Variable guidance methods] (i.e., E-CFG, β-CFG, RAAG, FDG): improve quality but *no computation reduction*.

OUSAC jointly optimizes on both directions, **achieving reduced computation while maintaining or even improving quality**. We enable (1) fewer total steps through optimized guidance scales, and (2) fewer CFG steps by adjusting scales at certain timesteps to compensate for skipping CFG at others. This creates a search space ($2^T$ patterns × continuous scales) too large for hand-crafted design, motivating our proposed evolutionary optimization.


For the 2nd stage caching, OUSAC is the **first to identify** that variable guidance breaks standard caching assumptions (~40% higher error), motivating the novel **adaptive rank allocation** design. We also discover cross-scale generalization via $k \cdot w^*$, eliminating the need for per-scale re-optimization.

---

### Meta-Review · Area_Chair_ox8s · 2026-01-06

**Summary:**

Reviewers raised concerns about the paper’s incremental novelty, limited theoretical insight, and reliance on engineering-heavy components.

**Reviewer Concerns:**

The rebuttal adequately addressed concerns about evaluation on modern models (e.g., FLUX), generalization across prompts/scales, and optimization cost.
No reviewer participate in discussion and these concerns raised by reviewers: the reliance on evolutionary search—though feasible—is still engineering-heavy with limited theoretical insight. Additionally, while new benchmarks were added, the core novelty of combining sparse CFG and caching remains incremental to some reviewers.

**Reviewer Scores:**

The reviewers did not participate in the discussion, so I think they may not modify the scores

---

### Decision · Program_Chairs · 2026-01-26

Reject